# Whi7 is an unstable cell-cycle repressor of the Start transcriptional program

Mercè Gomar-Alba[1], Ester Méndez[1], Inma Quilis[1], M. Carmen Bañó[1] & J. Carlos Igual[1]

Start is the main decision point in eukaryotic cell cycle in which cells commit to a new round of cell division. It involves the irreversible activation of a transcriptional program by G1 CDK-cyclin complexes through the inactivation of Start transcriptional repressors, Whi5 in yeast or Rb in mammals. Here we provide novel keys of how Whi7, a protein related at sequence level to Whi5, represses Start. Whi7 is an unstable protein, degraded by the SCF[Grr1] ubiquitin-ligase, whose stability is cell cycle regulated by CDK1 phosphorylation. Importantly, Whi7 associates to G1/S gene promoters in late G1 acting as a repressor of SBF-dependent transcription. Our results demonstrate that Whi7 is a genuine paralog of Whi5. In fact, both proteins collaborate in Start repression bringing to light that yeast cells, as occurs in mammalian cells, rely on the combined action of multiple transcriptional repressors to block Start transition.

[1] Departament de Bioquímica i Biologia Molecular and Estructura de Recerca Interdisciplinar en Biotecnologia i Biomedicina (ERI BIOTECMED), Universitat de València, Burjassot, 46100 Valencia, Spain. Correspondence and requests for materials should be addressed to J.C.I. (email: jcigual@uv.es)

Cell cycle progression is controlled by a sophisticated regulatory system in which integrated networks of switch-like mechanisms help to organize an ordered succession of distinct cyclin-dependent kinase (CDK) activities that trigger the different cell cycle events[1]. Two fundamental molecular processes are at the core of the cell cycle control system: gene expression and protein degradation. They govern the temporally and orderly accumulation of key cell cycle regulators and many other proteins required for cell cycle events.

The major point in cell cycle control occurs at the end of G1 phase in a process called Start in yeast and Restriction Point in mammalian cells[2]. At this stage, cells decide to initiate or not a new round of cell division. Molecular strategy is extraordinarily well conserved between yeast and mammals. It involves the activation by specific CDK kinases of a transcriptional program involving hundred of genes[3], which provides the coherent expression of key cell cycle regulators and the cellular machineries required for the early events of the cell cycle. Failure to proper regulate cell cycle entry can result in abnormal division and lead to cancer[4].

In *Saccharomyces cerevisiae*, Start transcriptional gene expression depends on two related transcription factors: SBF and MBF[5, 6]. Both are heterodimeric complexes constituted by a common regulatory subunit, Swi6, and a different but related sequence-specific DNA binding protein, Swi4 in SBF and Mbp1 in MBF. SBF controls expression, among others, of *CLN1* and *CLN2* G1 cyclin genes and genes encoding proteins involved in morphogenesis. MBF regulates periodic expression of genes involved in DNA metabolism and *CLB5* and *CLB6* S-phase cyclin genes. Although each factor preferentially regulates specific genes, they show significant functional overlap[7, 8]. This redundancy and the importance of this transcription program are emphasized by the lethality of *swi4swi6* and *swi4mbp1* double mutants[9].

Transcriptional activation at G1/S is regulated by the G1 CDK-cyclin activities (Cdc28 associated with G1 cyclins Cln1, Cln2, and Cln3 in the case of *S. cerevisiae*). It follows a complex ordered series of events. SBF binds to target promoters in early G1, but due to the binding of Whi5 transcriptional repressor, it does not initiate transcription until late in G1[10, 11]. Activation of Start gene expression is initially promoted by the Cln3-Cdc28 kinase[12, 13]. In early G1, Cln3-Cdc28 is attached to the endoplasmic reticulum membrane (ER)[14]. Later on in G1, the chaperone Ydj1 releases Cln3-Cdc28 from the reticulum, allowing its nuclear accumulation[15]. Then, Cln3-Cdc28 promotes the dissociation of Whi5[16, 17], triggering transcription. As a result, kinases Cln1,2-Cdc28 accumulate and act on Whi5, SBF, and MBF to strengthen a genome-wide transcriptional response and to establish a positive feedback loop that drives cellular commitment and gives coherence to G1-S transition[18–20].

Different scenarios have been recently proposed on how activation of the transcriptional program could be integrated with cell growth. They included the dilution of Whi5 due to cell growth[21],

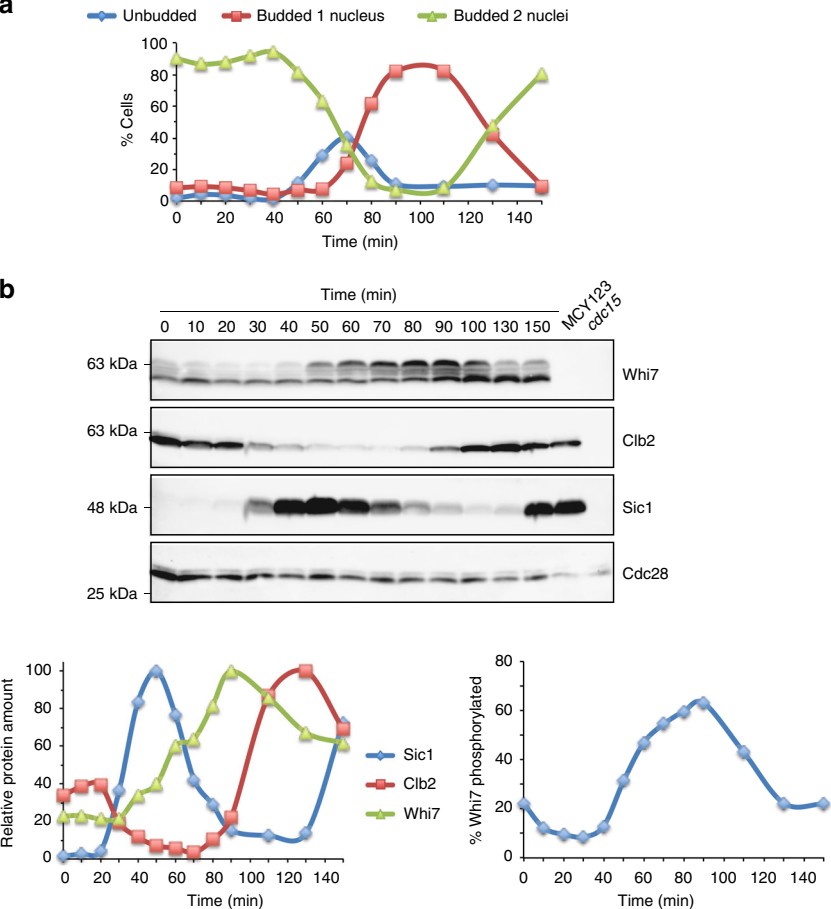

**Fig. 1** Cell cycle regulation of Whi7. *cdc15* cells expressing GFP-tagged Whi7, HA-tagged Clb2, and myc-tagged Sic1 (JCY1802) were arrested in telophase by incubation at 37 °C. After 3 h, cells were transferred to 25 °C and cell cycle progression was analyzed at the indicated time. **a** Graph shows the distribution of cells at different cell cycle stages. **b** Whi7, Clb2, and Sic1 proteins, as well as Cdc28 as loading control, were analyzed by western blot; *left graph* shows the relative amount of each protein referred to the most abundant sample; *right graph* represents the percent of total Whi7 protein present as slow migrating bands. MCY123 and no tagged *cdc15* strains were included in western analysis as controls

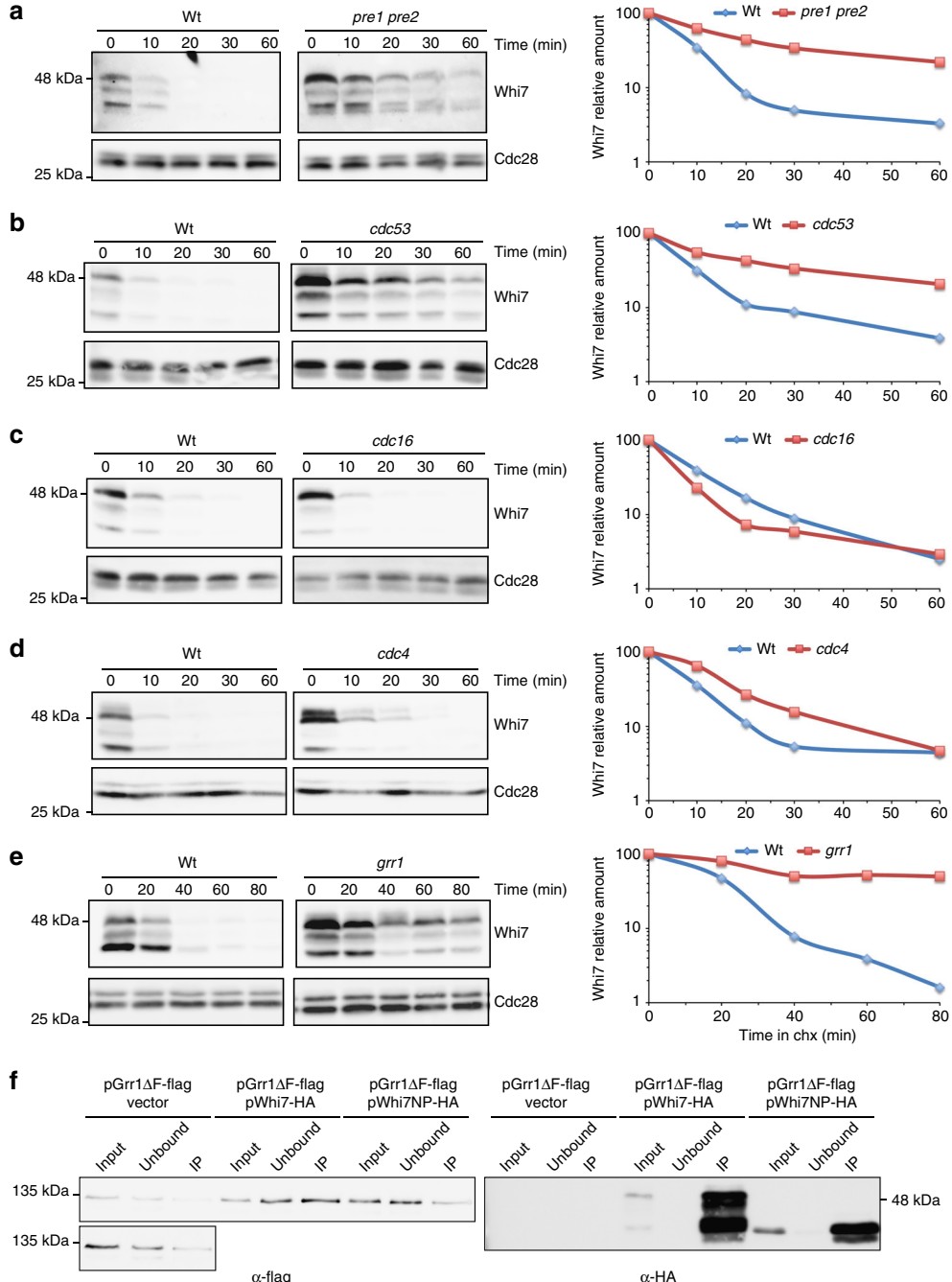

**Fig. 2** Analysis of Whi7 degradation in ubiquitin-ligase mutant strains. **a–e** Cells of the *pre1 pre2* (JCY1740) and its parental WCG4α (JCY1739), *cdc53* (JCY1732), *grr1* (JCY1760), *cdc4* (JCY1757) and their parental W303 (JCY1728), and *cdc16* (JCY1737) and its parental A364 (JCY1735) strains expressing a HA-epitope-tagged Whi7 protein at endogenous level were transferred at 37 °C for 3 h (except the *grr1* mutant and its parental strain) and then incubated in the presence of 100 μg mL⁻¹ cycloheximide (chx). Whi7 protein level was analyzed at the indicated time after the addition of cycloheximide by western blot. Cdc28 is shown as loading control. Graph represents the relative amount of Whi7 protein related to Cdc28. **f** The *grr1* mutant strain (JCY1760) was co-transformed with a plasmid expressing a flag-tagged Grr1 lacking its F-box (Grr1ΔF) and either the pWHI7 or pWHI7-NP plasmid (expressing HA-tagged wild-type or a non-phosphorylatable Whi7 respectively) or a control vector. Cells were grown on raffinose medium and transferred to galactose medium for 4 h. Whi7 was immunoprecipitated (IP) from crude extracts and the presence of Grr1ΔF and Whi7 in the input, unbound and immunoprecipitated fractions was determined by western analysis. A longer exposition of the vector samples is shown for better evaluate background signal

the competition between cell growth and cell cycle entry for the Ydj1 chaperone[22] and the titration of increasing amounts of Cln3 molecules due to cell growth against the constant number of SBF binding sites in DNA[23].

Later in the cell cycle, both SBF and MBF dependent transcription are silenced by different mechanisms. While SBF is inactivated by phosphorylation by Clb-Cdc28 kinases[24], transcriptional repressor Nrm1 acts via a negative feedback to inactivate MBF after G1/S transition[8]. In addition to Whi5 and Nrm1, other proteins associate with SBF and MBF. Msa1 and Msa2 bind both factors[25]; playing a role in response to stress or in the transition to quiescence[26, 27]. The Swi6-interacting protein Stb1 is

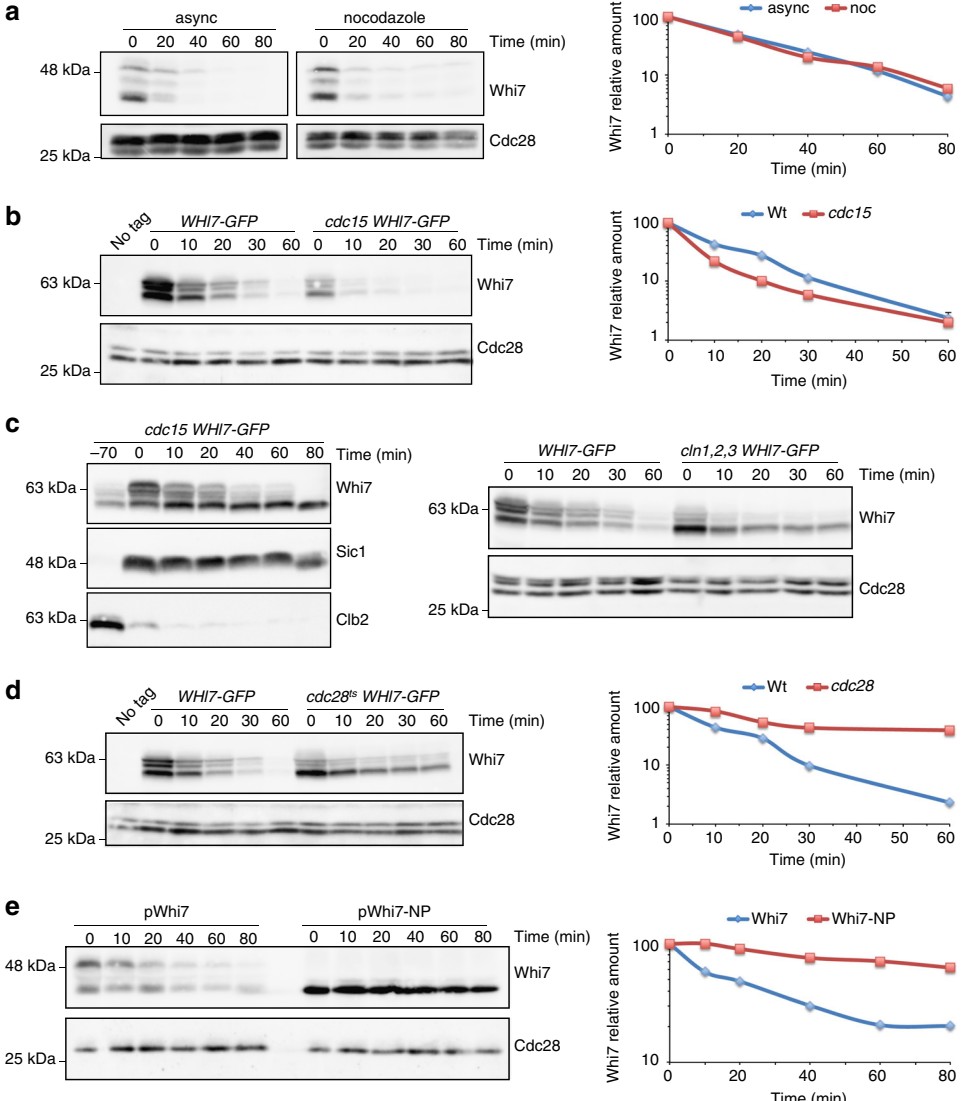

**Fig. 3** Cell cycle regulation of Whi7 protein stability by Cdc28 phosphorylation. **a** Cells of the *WHI7-HA* (JCY1728) strain were incubated for 2 h in the presence of 15 μg mL$^{-1}$ nocodazole (>90% of cells arrested at G2/M phase). Whi7 protein stability was analyzed as described in Fig. 2. **b** Wild-type (JCY1746) and *cdc15* mutant (JCY1802) cells expressing GFP-tagged Whi7 were incubated at 37 °C for 3 h (>95% of *cdc15* mutant cells arrested at telophase), and then Whi7 protein stability was analyzed. **c** *left panel: cdc15* cells expressing GFP-tagged Whi7 (JCY1802) were arrested in telophase by incubation at 37 °C. After 3 h, cells were released from the arrest and after 50 min, 100 μg mL$^{-1}$ cycloheximide was added. Whi7, as well as Clb2 and Sic1 as controls of cell cycle progression, were analyzed. No budding occurred during the experiment confirming that cells remain in G1 phase. *right panel*: wild type (JCY1746) and *GAL1:CLN3 cln1 cln2* (JCY2008) cells expressing GFP-tagged Whi7 were grown on galactose, transferred to YPD medium and after 3 h (100% of cell in G1 in the mutant strain) Whi7 protein stability was analyzed. **d** Wild type (JCY1746) and *cdc28* mutant (JCY1789) cells expressing GFP-tagged Whi7 were incubated at 37 °C for 3 h and then Whi7 protein stability was analyzed. **e** Whi7 protein stability in *whi7* mutant cells (JCY1819) transformed with a centromeric plasmid that expresses the HA-tagged wild type (pWHI7) or a mutated protein in all consensus CDK phosphorylation sites (pWhi7-NP)

required for efficient repression previous to Start[23, 28] and sharp periodic expression preferentially in MBF regulated genes[29, 30]. Also, the Pcl9-Pho85 CDK associates to promoters and through Whi5 phosphorylation collaborates with Cln-Cdc28 in transcriptional activation[31].

The second major mechanism affecting the levels of cell cycle regulators is proteolysis mediated by ubiquitination and degradation in the proteasome. Two ubiquitin ligases are important in cell cycle regulation: SCF, critical for G1/S transition, and APC, essential in mitosis[32–35]. SCF consists of four subunits: Skp1, Cdc53, Rbx1, and a F-box protein which is responsible for substrate recognition. Two F-proteins are involved in the degradation of most cell cycle regulators: Grr1 mediates degradation of G1

Cln cyclins whereas Cdc4 controls CKI Sic1 degradation. However, some overlapping sets of substrates may exist[36]. Recognition of substrate by SCF requires the phosphorylation of specific residues[35, 37].

Whi7/Srl3 was originally identified as a multicopy suppressor of *rad53* lethality[38]. It constitutes together with Whi5 and Nrm1 a family protein characterized by the presence of the GTB (G1/S transcription factor binding) motif. In the case of Whi5 and Nrm1, this motif mediates transcriptional repression binding to SBF or MBF, respectively[39]. Unlike *whi5* mutant, *whi7* does not present a small cell size phenotype in asynchronous cultures, apparently discarding a role in cell cycle initiation[10, 40]. However, very recently Whi7 has been connected to Start regulation at its

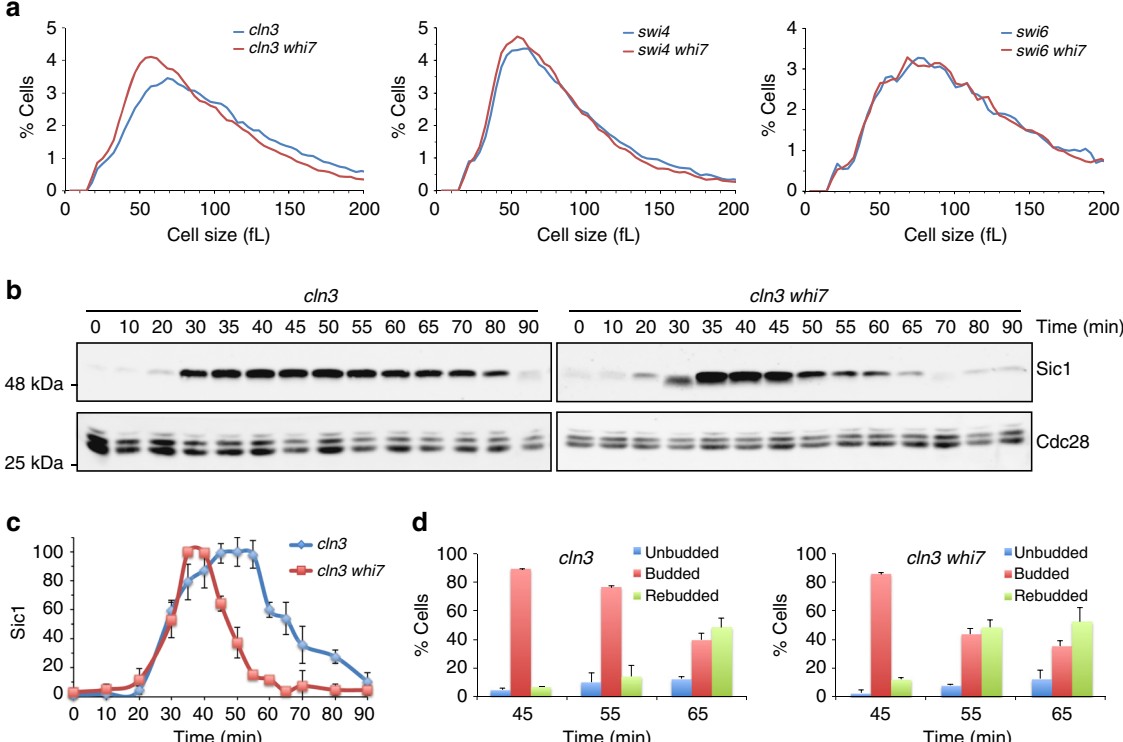

**Fig. 4** Effect of *whi7* mutation in Start timing in a *cln3* mutant strain. **a** Cell size distribution in cultures of the *cln3* (MT244), *cln3 whi7* (JCY1868), *swi4* (JCY167), *swi4 whi7* (JCY1872), *swi6* (JCY221), and *swi6 whi7* (JCY1880) mutant strains. **b** Cultures of *cdc15 cln3 SIC1-myc* (JCY1837) and its derivative *whi7* (JCY1896) mutant strain were arrested in telophase by incubation at 37 °C. After 3 h, cells were transferred to 25 °C and cell cycle progression was analyzed at the indicated time. Western blot shows Sic1 and Cdc28 (as loading control) protein level. **c** Plot represents the relative amount of Sic1 protein relative to Cdc28. **d** Budding as a marker of cell cycle initiation was monitored at the indicated times. Note that *cdc15* mutant often shows a delay in cell separation after the release from the arrest, giving rise to the apparition of rebudded cells. Values in **c** and **d** are the mean and s.d. derived from three experiments

earliest steps. Concretely, Whi7 helps to retain Cln3 in the ER membrane, a function that is inhibited by CDK dependent phosphorylation[41].

Here we show that Whi7 plays a new role in Start regulation independent from the control of Cln3 localization. Our results demonstrate that Whi7 is an unstable cell cycle regulated protein that acts as a genuine paralog of Whi5 repressing the Start transcriptional program.

## Results

**Whi7 level and phosphorylation is cell cycle regulated.** Many cell cycle regulators are periodic proteins whose level fluctuates through the cell cycle. Because of that, we carried out an analysis of Whi7 protein in synchronized cultures. After release from a telophase arrest induced by a thermosensitive *cdc15* mutation, progression through the cell cycle was analyzed by the presence of bud and number of nuclei (Fig. 1a) and the level of mitotic Clb2 cyclin or CKI Sic1 proteins (Fig. 1b). Clb2 decay and Sic1 accumulation reflected mitotic exit at approximately 40 min; later on, Sic1 degradation and budding marked the execution of Start at ~60 min; the appearance of Clb2 marked the G2 phase at 90 min, whereas progression through anaphase was revealed by the increase in cells with segregated nuclei at 130 min. As it is observed in Fig. 1b, Whi7 migrates in SDS-PAGE as multiple bands, which correspond to distinct phosphorylated states since lambda phosphatase treatment resulted in the migration as a single band of higher mobility (Supplementary Fig. 1). Whi7 level oscillates along the cell cycle, increasing in early G1 before Start, peaking in G2 and decaying in mitosis. Importantly, changes in Whi7 phosphorylation along the cell cycle were also observed,

Whi7 becoming hyperphosphorylated as cells progress from early G1 to mitosis. In conclusion, our results revealed that Whi7 is cell cycle regulated and suggest two states along the cell cycle: one associated with hyperphosphorylation and higher protein level from early G1 to G2/M, and a second one associated with hypophosphorylation and lower protein level in M/early G1.

**Whi7 is an unstable protein degraded mainly via SCF^{Grr1}.** Many cell cycle regulators are unstable proteins. Therefore, we analyzed Whi7 protein stability by translational shut-off experiments. The result indicated that Whi7 is indeed an unstable protein with a half-life of approximately 20 min at 25 °C and <10 min at 37 °C (Supplementary Fig. 2).

Whi7 was stabilised in the *pre1 pre2* proteasome mutant strain (Fig. 2a), indicating that degradation of Whi7 is accomplished by the ubiquitin-proteosome pathway. Two ubiquitin ligases are involved in cell cycle regulation: SCF and APC. Whi7 stability was analyzed in mutant strains in either the SCF subunit Cdc53 or the APC subunit Cdc16. As it can be observed in Fig. 2b, c, while Whi7 remains highly unstable in *cdc16* mutant cells, it was highly stabilised when Cdc53 was inactivated. This indicates that the ubiquitin-ligase SCF mediates Whi7 degradation.

SCF recognizes specific cell cycle substrates mainly by means of two F-protein subunits: Cdc4 and Grr1. Whi7 still showed a high instability in the absence of Cdc4, although somewhat lower to that observed in the wild-type strain (Fig. 2d). On the contrary, Whi7 was highly stabilized in the *grr1* mutant cells (Fig. 2e). This indicates that Grr1 is the major F-protein involved in Whi7 degradation. In the case of Cln3, which is also degraded by Grr1, it has been demonstrated that the insensitivity to Cdc4 is due to

Cdc4 nuclear localization instead of a genuine Cdc4 inability to recognize Cln3[36]. However, no changes in Whi7 stability were observed when nuclear export of Cdc4 was forced by the fusion of a nuclear export signal (Cdc4-NES), confirming a residual role for Cdc4 in Whi7 degradation (Supplementary Fig. 3). Finally, we detected a physical interaction between Whi7 and Grr1 (Fig. 2f). In conclusion, all these results revealed that Whi7 is an unstable protein, degraded mostly via the SCF[Grr1] ubiquitin ligase.

**Whi7 stability is cell cycle regulated by Cdc28 phosphorylation.** Stability of cell cycle regulators fluctuates along the cell cycle. To test whether Whi7 degradation could be cell cycle regulated, protein stability was studied at different cell cycle stages. When cells were blocked in metaphase after incubation with nocodazole,

Whi7 showed a short half-life similar to that observed in asynchronic cultures (Fig. 3a). Whi7 was also highly unstable in cells arrested in telophase in a *cdc15* mutant strain (Fig. 3b). To test stability in G1 cells, cycloheximide was added 50 min after release from a *cdc15*-induced arrest; the absence of budding and the presence of Sic1 confirmed that cells remain in G1 along the experiment. On the contrary to that observed in G2/M and telophase, Whi7 remained stable in G1 cells. The same result was observed in G1 cells obtained by depletion of Cln cyclins (Fig. 3c).

It is noteworthy that the stable Whi7 protein appears in the western blot as the non-phosphorylated band, strongly suggesting that Whi7 stability is controlled by phosphorylation. This is consistent with the observation that Whi7 degradation depends on the SCF pathway, which requires phosphorylation of target

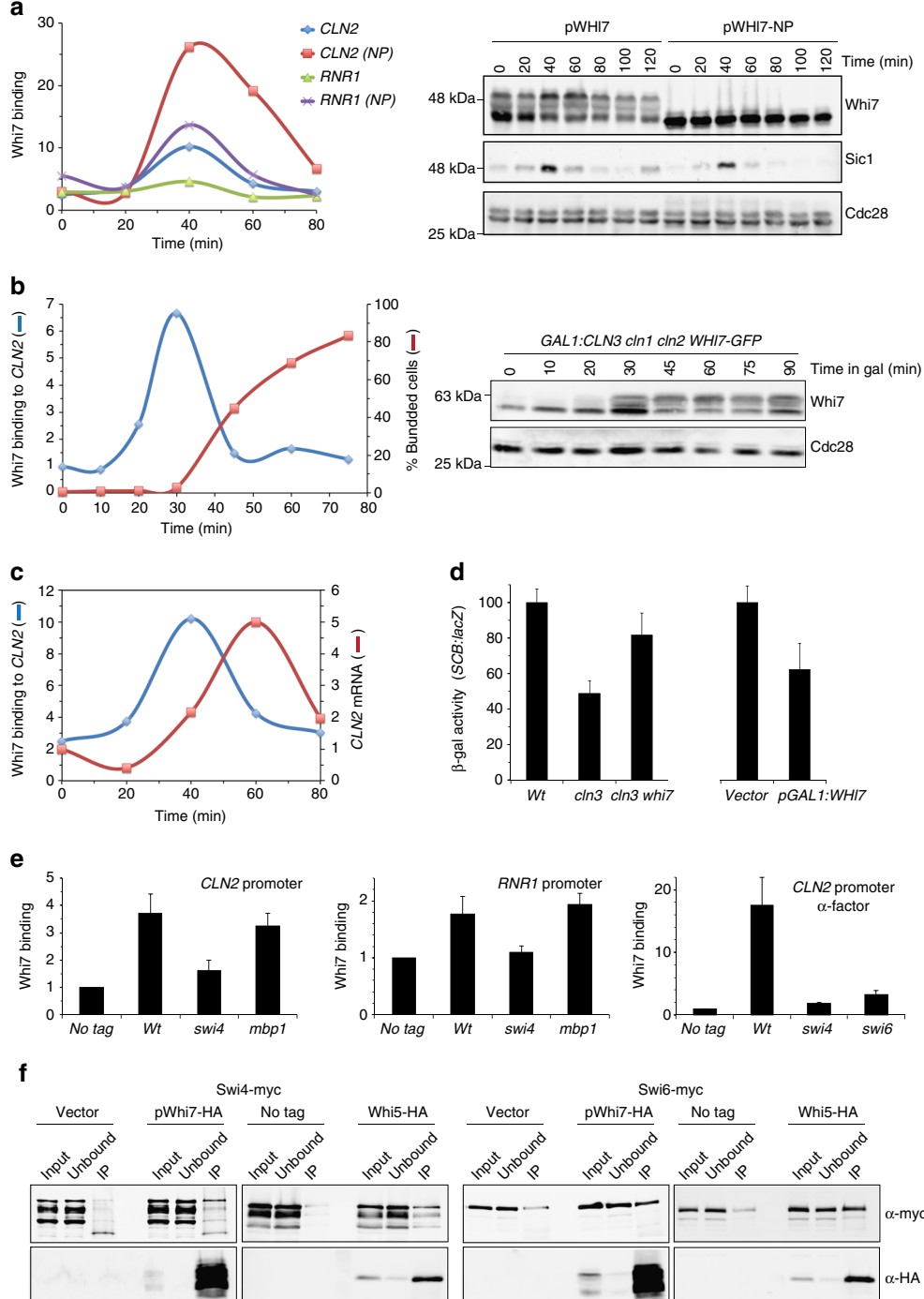

proteins. The CDK Cdc28 is a good candidate to mediate phosphorylation-dependent degradation of Whi7. We observed that Whi7 was stabilized in *cdc28* mutant cells, accumulating in a non-phosphorylated form (Fig. 3d). Moreover, Whi7-NP (a Whi7 variant with all the Cdc28 phosphorylation consensus sites mutated) remains totally stable (Fig. 3e). Consistently, interaction with Grr1 is loss in the non-phosphorylatable Whi7 protein (Fig. 2f).

To date, Cln3 is the only cyclin known to drive Whi7 phosphorylation. The fact that Whi7 was phosphorylated and unstable in G2/M and telophase arrested cells, suggests that other cyclins are involved. In fact, no differences in Whi7 mobility pattern and stability were observed in *cln3* when compared to wild-type cells, which demonstrated that other cyclins distinct from Cln3 are also responsible for Whi7 phosphorylation and degradation (Supplementary Fig. 4a). Supporting this, Whi7 protein was partially stabilized in a mutant strain in Clb1-4 cyclins (Supplementary Fig. 4b) and after inactivation of Cln1 and Cln2 in *cdc4* mutant cells (Supplementary Fig. 4c).

In summary, our results indicated that Whi7 is an unstable protein whose degradation is cell cycle regulated by Cdc28-dependent phosphorylation: the protein is stable only in G1, the period of the cell cycle with no CDK activity.

**Whi7 has a Cln3-independent function in repressing Start**. We wonder whether Start regulation by Whi7 could involve other mechanisms distinct from the described Whi7 role in Cln3 cytosolic retention[41]. In a wild-type background, inactivation of Whi7 caused a significant decrease in the percentage of unbudded cells in asynchronous cultures: $18.9 \pm 2.7$ in *whi7* compared to $24.3 \pm 2.4$ in wild type (budding index values are mean and s.d. derived from at least five independent culture henceforth). This is in agreement with the reported decrease in cell size at budding[41] indicating a shortened G1 phase. However, this effect did not alter the cell size distribution in asynchronous cultures, probably due to compensation in other cell cycle stages (Supplementary Fig. 5). In the case of *cln3* mutant cells, which show an increased cell size due to a defective Start activation, a reduction in the percentage of unbudded cells was also evident when Whi7 was inactivated ($31.1 \pm 2.4$ compared to $40.5 \pm 3.1$). Importantly, in this case *WHI7* deletion manifested a mild, but significant, reduction in cell size in asynchronous cultures (Fig. 4a). This result suggests that Whi7 mediates some Start function independently from Cln3.

To examine the influence of Whi7 on the timing of G1/S transition, G1 phase length was investigated in *cln3* mutant cells. After release from a *cdc15*-induced telophase arrest, both *WHI7* and *whi7* cells exit from mitosis (marked by Sic1 accumulation) with similar kinetics. However, Start execution (marked by Sic1

disappearance and budding) was advanced ~10 min in the absence of Whi7 (Fig. 4b–d and Supplementary Fig. 6). These observations clearly demonstrate that Whi7 represses Start progression by a mechanism that is independent from Cln3 regulation.

**Whi7 associates with Start gene promoters in G1**. Whi7 is related to Whi5 at sequence level. Therefore, it is tempting to speculate the possibility that Whi7 could play a new role in cell cycle control related to transcriptional regulation. We investigated the association of Whi7 to genes of the G1/S transcriptional program by chromatin immunoprecipitation (ChIP). Cells were synchronized by telophase arrest and release, and the specific purification of DNA fragments from *CLN2* and *RNR1* promoters in immunoprecipitated Whi7 samples was assayed. As it is observed in Fig. 5a, a transient binding of Whi7 to both *CLN2* and in a lesser extent to *RNR1* promoter was detected after the release, peaking at the G1 phase, as deduced by the absence of budding and the presence of Sic1. Concomitant with Start execution (budding and Sic1 degradation), Whi7 binding decays. Whi7 association to *CLN2* promoter was also detected with a genomic GFP-tagged *WHI7* (Supplementary Fig. 7). Association to Start genes was also observed when cells express the non-phosphorylatable Whi7-NP variant, indicating that phosphorylation by Cdc28 is not required for binding.

To better characterize Whi7 binding during G1, we used a G1-arrest induced by Cln cyclins depletion. Whi7 is not found associated to *CLN2* promoter in arrested cells but, a transient binding before Start was observed after release from the arrest (Fig. 5b). In conclusion, Whi7 associates with the G1/S gene promoters in late G1 and is released at Start; this resembles the action of Whi5 and clearly pointed to a new role for Whi7 in Start regulation through the control of the Start transcriptional program.

To evaluate the relationship between Whi7 binding to promoters and Start transcriptional activation, we analyzed in parallel *CLN2* gene expression. Whi7 dissociation correlated with *CLN2* transcriptional activation (Fig. 5c). Moreover, deletion of *WHI7* overcame, although partially, the defective SBF driven transcription of a *cln3* mutant strain, and overexpression of *WHI7* reduced expression of the SBF-regulated *SCB:lacZ* reporter (Fig. 5d). All these observations support a role of Whi7 as a transcriptional repressor of G1/S specific genes.

**Whi7 binds to the SBF transcription factor**. The Start transcriptional program is mediated by the related SBF and MBF transcriptional factors. To test whether Whi7 acts through one of them, the binding of Whi7 to *CLN2* (primarily regulated by SBF) and *RNR1* (primarily regulated by MBF) was investigated in

**Fig. 5** Analysis of Whi7 function in transcription at Start. **a** Cultures of *cdc15 SIC1-myc whi7* (JCY1843) transformed with the pWHI7 or pWHI7-NP centromeric plasmid were arrested in telophase by incubation for 3 h at 37 °C (>95% of cells arrested). After the release from the arrest, Whi7 binding to *CLN2* and *RNR1* promoter was investigated by ChIP assays. Western blot shows Whi7, Sic1, and Cdc28 (as loading control). Cell cycle progression was monitored by the oscillations of Sic1 and bud emergence that indicated execution of Start at 60 min. **b** Cultures of *GAL1:CLN3 cln1 cln2 WHI7-GFP* (JCY2008) were arrested in G1 by incubation for 3 h in YPD medium. Then, cells were transferred to YPGal medium and Whi7 binding to *CLN2* promoter was investigated by ChIP assays. Western blot shows Whi7 and Cdc28. Cell cycle progression was monitored by bud emergence that indicated execution of Start at 45 min. **c** The level of *CLN2* mRNA relative to *ACT1* mRNA as a control was analyzed by quantitative RT-PCR in cells transformed with the pWHI7 plasmid. **d** β-galactosidase activity in extracts from wild type (W303), *cln3* (MT244), and *cln3 whi7* (JCY1868) strains (*left*) as well as from wild-type cells containing a control vector or the pGAL1:WHI7 plasmid (*right*), transformed with the pSCB:lacZ reporter gene. **e** Whi7 binding to *CLN2* and *RNR1* promoter was investigated by ChIP assays in asynchronous cultures of wild type (W303), *swi4* (JCY167), and *mbp1* (JCY624) and in α-factor arrested cultures of wild type (W303), *swi4* (JCY167), and *swi6* (JCY325) strains transformed with the pWHI7 plasmid or a control vector (no tag). **f** Cells expressing a myc-tagged Swi4 (JCY1879) or myc-tagged Swi6 (JCY622) proteins at endogenous level were transformed with the pWHI7 plasmid or a control vector. Whi7 was immunoprecipitated from crude extracts and the presence of Whi7, Swi4-myc, and Swi6-myc in the input, unbound, and immunoprecipitated fractions was determined by western analysis. The same experiment was carried out with cells expressing a HA-tagged Whi5 and either myc-tagged Swi4 (JCY1984) or myc-tagged Swi6 (JCY1982). Values in **d** and **e** are the mean and s.d. derived from at least three experiments

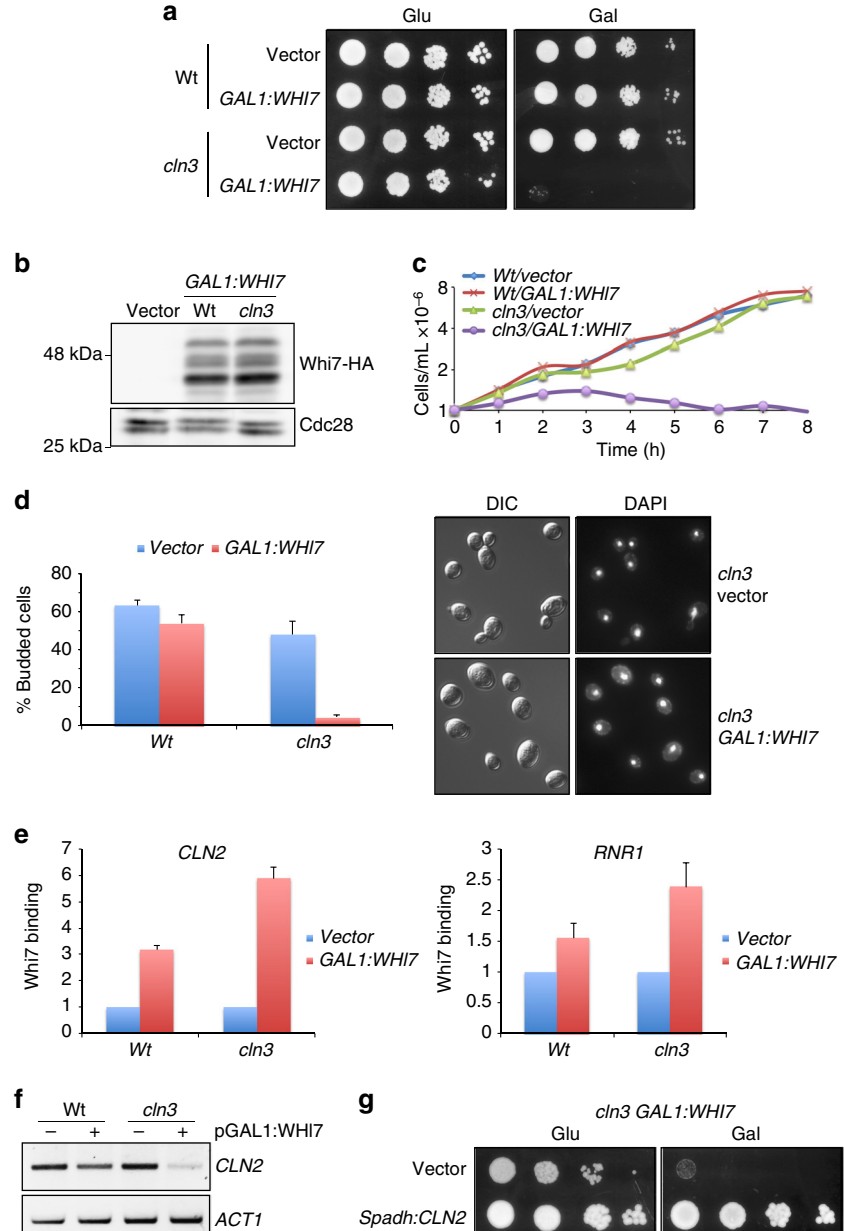

**Fig. 6** Effect of *WHI7* overexpression in cell cycle progression in a *cln3* mutant strain. **a** 10-fold serial dilutions from cultures of wild-type (W303) and *cln3* mutant (MT244) strains transformed with an empty vector or the pGAL1:WHI7 plasmid, were spotted onto SC-Glu and SC-Gal medium and incubated at 25 °C for 3 days. **b** Western blot analysis of Whi7 protein level in cells grown on SC-Gal medium. **c** The same strains were grown on SC-Raf. Galactose to 2% was added to induce overexpression of *WHI7* and the increase in cell number was analyzed. **d** Cell cycle distribution of cells after 3 h since the addition of galactose. Plot represents percentage of budded cells. Pictures show DIC image and DAPI staining of DNA. **e** Whi7 association with *CLN2* and *RNR1* promoters was investigated by ChIP assays after 3 h since the addition of galactose. **f** The level of *CLN2* and *ACT1* mRNA was analyzed by semiquantitative RT-PCR after 3 h since the addition of galactose. **g** 10-fold serial dilutions from cultures of the *cln3* mutant strain (MT244) transformed with pGAL1:WHI7 and either a vector or a plasmid expressing *CLN2* under the control of the *adh* promoter from *S. pombe*, were spotted onto SC-Glu and SC-Gal medium and incubated at 25 °C for 3 days. Values in **d** and **e** are the mean and s.d. derived from three experiments

mutant strains in the Swi4 and Mbp1, specific components of SBF and MBF, respectively. The results revealed that in the absence of Swi4, Whi7 association to both *CLN2* and *RNR1* was severely reduced whereas in the absence of Mbp1 there was a very slight or no effect (Fig. 5e). Moreover, binding of Whi7 to *CLN2* promoter was also severely affected by the absence of Swi6 (Fig. 5e). This result demonstrated that Whi7 binds Start gene promoters mostly in a SBF-dependent manner.

To further characterize the SBF-Whi7 connection, we carried out co-immunoprecipitation assays. As shown in Fig. 5f, both Swi6 and Swi4 selectively co-purified with immunoprecipitated

Whi7 in a range similar to that observed with Whi5. These results provide support for a specific physical interaction of Whi7 with Swi4 and Swi6 *in vivo*.

**Whi7 overexpression causes a G1 arrest in *cln3* mutant cells.** To further demonstrate a Whi7 repressor role independent from Cln3, we determined the effect of overexpressing *WHI7* from the *GAL1* promoter. In wild-type cells, there was no significant effect on cell growth but the percentage of unbudded cells in mid-log cultures raised from $32.2 \pm 2.1$ (vector) to $49.4 \pm 2.7$ (*GAL1*:

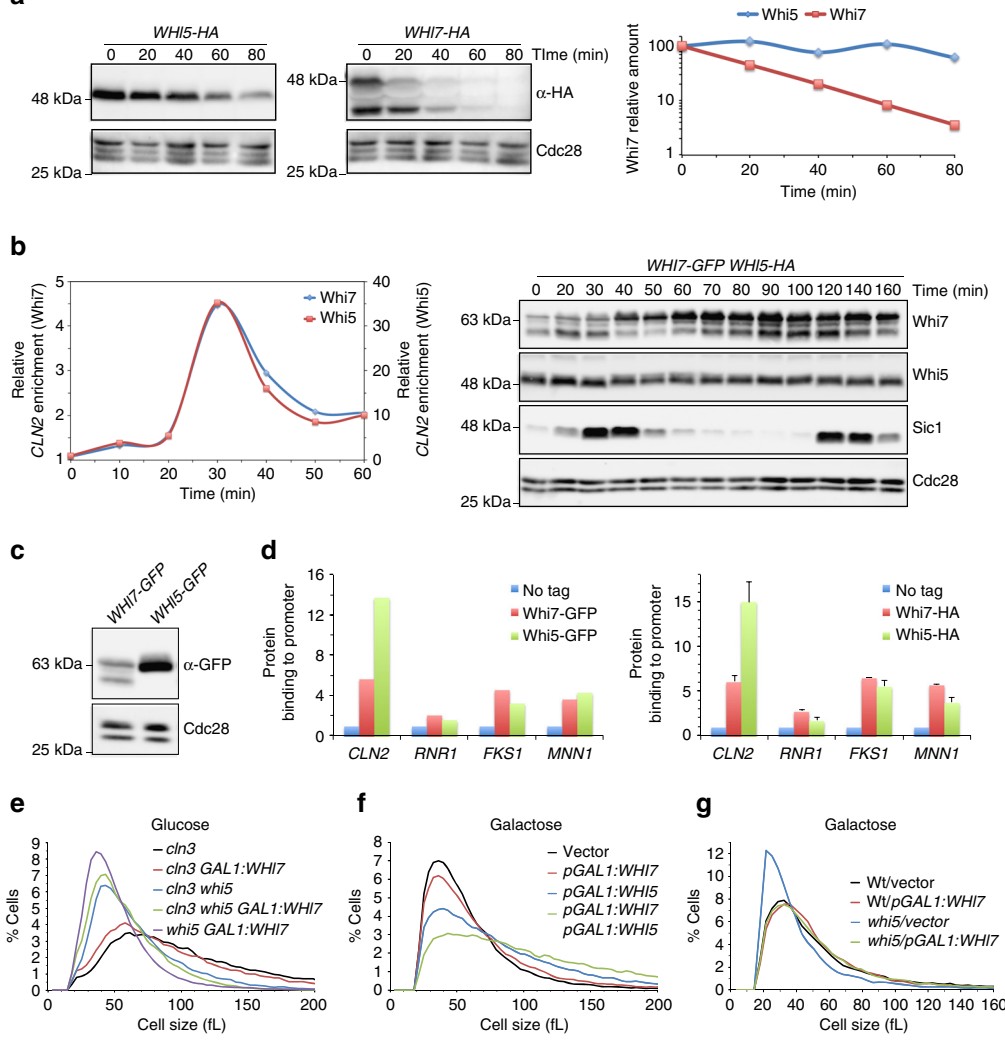

**Fig. 7** Comparative analysis of Whi7 and Whi5 proteins. **a** Protein stability in *WHI5-HA* (JCY734) and *WHI7-HA* (JCY1728) cells was analyzed as described in Fig. 2. **b** Exponentially growing cultures of *cdc15 SIC1-myc WHI5-HA WHI7-GFP* (JCY1912) were arrested in telophase by incubation for 3 h at 37 °C. After release from the arrest, Whi5 and Whi7 binding to *CLN2* promoter were investigated by ChIP assays. Western blot shows Whi5, Whi7, Sic1, and Cdc28 (as loading control). Cell cycle progression was monitored by the oscillations of Sic1 and bud emergence that indicated execution of Start at 40 min sample. **c** Analysis of protein level in wild-type cells expressing a GFP-tagged Whi7 (JCY1746) or Whi5 (JCY2009) protein. **d** Whi7 and Whi5 association to the indicated gene promoters was analyzed by ChIP assay in *GAL1:CLN3 cln1 cln2* cells expressing either Whi7-GFP (JCY2008) or Whi5-GFP (JCY2007) 30 min after release from the G1 arrest (*left panel*), and in exponentially growing cultures of wild-type cells expressing either Whi7-HA (pWHI7-HA plasmid) or Whi5-HA (JCY1346) (*right panel*; values are the mean and s.d. derived from three experiments). **e** Cell size distribution in cultures on YPD medium (*GAL1* promoter repressed to mimic *whi7* mutation) of *cln3* (MT244), *cln3 whi5* (JCY1875), *cln3 GAL1:WHI7* (JCY1883), *cln3 whi5 GAL1:WHI7* (JCY1921), and *whi5 GAL1:WHI7* (JCY1885) mutant strains. **f** Cell size distribution in cultures on SC-Gal medium of the wild-type (W303) strain transformed with either the pGAL1:WHI5, pGAL1:WHI7, both plasmids or the corresponding control vectors. **g** Cell size distribution in cultures on SC-Gal medium of the wild-type (W303) and *whi5* mutant (JCY1874) strains transformed with the pGAL1:WHI7 plasmid or a control vector

*WHI7* plasmid), which is in agreement with the reported increase in cell size at budding[41] indicative of an expanded G1 phase. Nevertheless, high levels of Whi7 were lethal in the absence of Cln3 (Fig. 6a). This is not due to differences in Whi7 expression between wild-type and *cln3* cells (Fig. 6b). Analysis of cells after *WHI7* induction indicated a first cycle arrest at Start with the accumulation of more than 97% of single-nucleated large unbudded cells (Fig. 6c, d).

Next, ChIP assays demonstrated an increased Whi7 association with *CLN2* and *RNR1* promoters in Start blocked *cln3* cells (Fig. 6e). This strongly suggests that Whi7 must cause G1 arrest by repressing the Start transcriptional program. In fact, analysis of *CLN2* mRNA in cells overexpressing Whi7 confirmed that gene expression was repressed (Fig. 6f). Moreover, ectopic

expression of *CLN2* supressed the lethality of *WHI7* over-expression in *cln3* cells, a typical property of mutants in the Start transcriptional program (Fig. 6g).

In conclusion, high levels of Whi7 caused repression of the Start transcriptional program leading to a G1 arrest in *cln3* mutant cells. This confirms that first, Whi7 plays a role in Start regulation mediating Start transcriptional program repression, and second, that this is a new role independent from its known function in Cln3 regulation.

**Whi7 function depends on the presence of SBF**. As described above, *WHI7* deletion causes a reduction in cell size in a strain genetically compromised for Start due to a *cln3* mutation.

Interestingly, *WHI7* deletion had no effect on cell size in the case of *swi4* or *swi6* mutation (Fig. 4a). This indicates that Whi7 effect on cell size depends on the presence of SBF but does not require Cln3.

The effect of *WHI7* overexpression in either *swi4* or *swi6* mutant cells was also investigated. It caused a reduction in cell growth rate and a small increase in the percentage of unbudded cells, in particular in the case of *swi4* cells, which is consistent with the known Whi7 regulation of Cln3 (Supplementary Fig. 8). Importantly, and contrary to what was observed in the case of *cln3* cells, high levels of Whi7 were not lethal in *swi4* and *swi6* mutant cells. This strongly suggests that Whi7 induced G1 arrest is mediated by SBF function.

**Whi7 is functionally redundant with the Start repressor Whi5.** Given the similarities between Whi7 functions described above with the role of the Start transcriptional repressor Whi5, we decided to compare both proteins. As described here, Whi7 is an unstable protein degraded by SCF^Grr1. Nevertheless, Whi5 is a pretty stable protein when compared to Whi7 (Fig. 7a).

We also investigated *CLN2* promoter association of Whi7 and Whi5 simultaneously in the same synchronized cells. Both proteins showed the same binding kinetics (Fig. 7b). This result also indicates that both factors are simultaneously occupying the *CLN2* promoter.

Using the same tagging cassette, we observed approximately a 50% reduced protein level of Whi7 compared to Whi5 (Fig. 7c). In parallel ChIP assays, binding of both proteins to different genes including *CLN2*, *RNR1* and SBF-regulated cell-wall genes *FKS1* and *MNN1* was detected in all cases, but differences in their specificity were consistently observed, Whi5 showing a preferred binding compared to Whi7 in the case of *CLN2* (Fig. 7d).

Next, we investigated whether inactivation of *WHI7* and *WHI5* have additive effect in cell size. Loss of *WHI7* does not affect the reduced cell size characteristic of a *whi5* mutant strain (Supplementary Fig. 5b). However, *WHI7* inactivation further reduces the cell size of a *whi5* mutant in a *cln3* background (Fig. 7e). This indicates that both proteins contribute to the control of cell size. It is noteworthy that *cln3 whi5 whi7* cells are bigger than *whi5 whi7* cells, which point to an additional function for Cln3 different from inactivating Whi5 and Whi7 in the control of Start. We also observed additive effect of Whi5 and Whi7 overexpression in the control of cell size, reinforcing the idea that both proteins act in parallel (Fig. 7f). Finally, we demonstrated that *WHI7* overexpression in *whi5* cells restores a normal cell size (Fig. 7g). This is an important observation that indicates that Whi7 protein is able to substitute Whi5 in Start regulation.

**Discussion**

Whi7 is related at sequence level to Whi5 transcriptional repressor, the yeast functional paralog of Rb protein in mammalian cells. This indicates a common evolutionary origin and suggests that both proteins could develop similar functions. In this work we provide new clues about Whi7 function in cell cycle regulation, characterizing Whi7 as an unstable cell-cycle regulated protein and bringing to light a novel function as a transcriptional repressor in Start.

The analysis of Whi7 protein along the cell cycle indicates that Whi7 is present in multiple phosphorylation states. Electrophoretic migration of Whi7 in a *cdc28* mutant and a Whi7 mutated in all the putative CDK phosphorylation sites was similar to that observed in lambda phosphatase assays, suggesting that CDK is the only kinase involved in Whi7 phosphorylation. However, it cannot completely be excluded that other kinases

may phosphorylate Whi7, as occurs for Whi5[17]. It was known that Whi7 is phosphorylated by Cln3-Cdc28 and consistently its phosphorylation pattern changed in G1/S[41]. Here, we found Whi7 in a phosphorylated state along basically the whole-cell cycle. Importantly, the degree of phosphorylation changes drastically. It begins to rise from mid-G1 prior to Start until mitosis, to decrease in mitotic exit. This temporal window matches the cell cycle period with CDK activity. Moreover, it strongly suggests that phosphatase Cdc14 may be responsible for Whi7 dephosphorylation, similarly to that reported for Whi5[42].

Cell cycle regulators are often unstable proteins whose degradation is regulated throughout the cycle. We have identified that Whi7 is degraded mainly via the ubiquitin-ligase SCF^Grr1. However, the fact that Whi7 is not completely stable in a *grr1* deletion mutant suggests that other pathways could contribute to its degradation or that Whi7 might have a high intrinsic instability. We also show that Whi7 degradation, as usually occurs in SCF targets, is dependent on CDK phosphorylation. This is consistent with the fact that Whi7 is stable only in G1, the period of low CDK activity. Cln3-Cdc28 is the first CDK activity appearing in the cycle, so it is reasonable to think that it initiates Whi7 degradation, but Whi7 is also unstable in G2/M or telophase and in *cln3* mutant cells and it is stabilized by the inactivation of *clb1-4* or *cln1-2*. Therefore, we propose that any CDK can phosphorylate Whi7 targeting it for degradation via SCF^Grr1, as occurs with other key cell cycle regulators such as the CKI Sic1[37] or the APC ubiquitin-ligase co-activator Cdh1[33].

Recently, Whi7 had been connected to cell cycle regulation based on the control of Cln3 cytosolic retention[41]. Here we describe that Whi7 inactivation produces a significant reduction in cell size and an advance in Start in *cln3* mutant cells. This is an important point because first, it indicates that Whi7 plays a key role as a negative Start regulator, and second, it highlights a new function distinct from its known role as regulator of Cln3. Whi7 showed sequence similarities with the Start regulator Whi5 but to date, no transcriptional function had been described for Whi7. Our results provide at last the demonstration that Whi7 is able to develop the same function as Whi5, binding and repressing the promoters of G1/S regulated genes. Several results support this idea: Whi7 associates with *CLN2* and *RNR1* promoters in G1 and its dissociation coincides with expression of G1/S genes, as Whi5 does; Whi7 inactivation overpass a defect in the expression of the SCB:lacZ reporter in *cln3* mutant cells and its overexpression is able to partially repress SCB:lacZ expression in wild-type cells; *WHI7* overexpression is lethal in a *cln3* mutant causing an arrest at Start because it remains attached to the promoters and maintains transcription locked; finally, Whi7 overexpression recovers normal cell size in *whi5* mutant cells clearly demonstrating that Whi7 is capable of performing the same function as Whi5.

Periodic expression in G1/S is mediated by SBF and MBF. Our results indicate that Whi7 binds to SBF regulated (*CLN2*) as well as to MBF regulated (*RNR1*) genes. However, Whi7 binding is not affected by Mbp1 mutation but is impaired by Swi4 or Swi6 mutation, confirming that binding to both *CLN2* as *RNR1* promoters is mediated by SBF. This is consistent with the detection of a lower Whi7 binding in *RNR1* than in *CLN2*. In addition, it has to be considered that SBF is indeed able to bind to *RNR1* promoter[8, 43, 44].

Overall, our results provide new insights in the characterization of Start control by demonstrating that Whi7 is, as Whi5, a repressor of the Start transcriptional program. Therefore, we propose Whi7 to be a functional paralog of Whi5. This deepens our knowledge of how Whi7 controls cell cycle. Thus, Whi7 plays a double repressive role in the execution of Start mediating: (1) the retention of Cln3 in the ER as previously described, and (2)

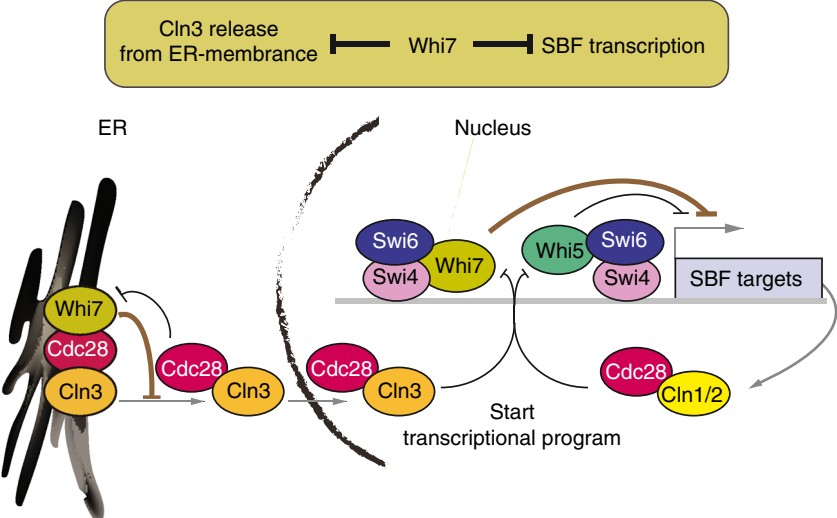

**Fig. 8** Model of Whi7 function in the control of Start. Whi7 plays a double repressive role in the execution of Start: it mediates the retention of Cln3 in the endoplasmic reticulum and it collaborates with Whi5 in the repression of SBF-dependent transcription

the repression at target promoters of Cln3 initial trigger and Cln1,2 feedback activation of SBF-dependent transcription (Fig. 8). This two-way lock mechanism highlights the necessity for robust controls in cell cycle regulation.

An open question is the role of Whi7 phosphorylation in this new function in transcription. A non-phosphorylatable version of Whi7 associates and dissociates from promoters normally. In the case of Whi5, mutations of all phosphorylation sites have no overt effect on Start and only when combined with non-phosphorylatable Swi6 are deleterious[10, 17]. Disengagement of Whi7 from SBF could also proceed via multiple redundant phosphorylation events in Whi7, Swi6, and Swi4. Supporting that Whi7 phosphorylation could help to release Whi7 from promoters, we observe largest promoter association with Whi7-NP compared to wild-type protein.

The fact that Whi7 overexpression restores normal cell size in a *whi5* mutant clearly reflects that Whi7 is able to perform the same function as Whi5. But what is the relevance of Whi7 in Start compared to Whi5? While Whi5 inactivation causes a dramatic reduction in cell size, Whi7 inactivation does not alter size distribution of wild-type or *whi5* cells. Only in a *cln3* mutant background we detect a cell size decrease when Whi7 is inactivated, and this effect is smaller than that observed by inactivating Whi5. Also, overexpression of Whi5 but not Whi7 caused a severe increase in cell size. Thus, the effect on cell size indicates that Whi5 is more critical than Whi7 for the correct timing of Start. In fact, ChIP assays revealed a 3–4 times more enrichment of *CLN2* promoter fragment with Whi5 than Whi7, which could suggest a greater presence of Whi5 than Whi7 in the *CLN2* promoter. However, it is important to note that we have detected simultaneous binding of Whi7 and Whi5 to *CLN2* promoter in the same cells, indicating that Whi7 is indeed involved in the regulation of Start transcription. Furthermore, Whi7 and Whi5 have additive effects on cell size control. Therefore, we propose a model in which the SCB sites of target promoters are occupied by SBF repressed largely by Whi5 but also by Whi7.

The presence of two repressor proteins should represent an evolutionary advantage to the cell. Functional redundancy between proteins is a hallmark in cell cycle regulation. Moreover, there could be a functional specialization of Whi7 and Whi5. Whi7 might be participating in a subset of genes and/or conditions. Whi7, but not Whi5, protein levels are specifically increased in mother cells after twelve generation[45], which could point to a

specific role for Whi7 in restraining cell cycle entry during replicative aging. Whi7 binds to *CLN2* promoter in the presence of hydroxyurea[39] and it was isolated as a suppressor of DNA-damage checkpoint *rad53* mutant[38], suggesting a function in the response to genotoxic stress. Furthermore, *WHI7* gene expression is regulated by the cell-wall integrity (CWI) pathway[46, 47] and it is known that SBF participates in gene expression of cell-wall genes[48, 49] and the CWI pathway regulates Swi4 and Swi6[50, 51]. Thus, cells might use Whi7 for transient responses under certain stress conditions.

The similarity in the regulatory control system of G1/S transition between yeast and mammalian are streaking. In mammalian cells, passage through the restriction point involves the turn-on of a wave of gene expression controlled by the E2F-DP family of transcription factors[3]. Until late G1, transcription is repressed by the association of Retinoblastoma family proteins (Rb, p107, p130) to E2Fs. Initial activation is triggered by cyclin D-CDK4,6 phosphorylation of Rb, which causes its dissociation from E2Fs. As a consequence, cyclin E gene is expressed, providing a positive feedback in which cyclin E-CDK2 phosphorylate Rb in order to maintain transcriptional activation. It is easy to recognize the same regulatory strategy in Start control in yeast, where the roles of cyclin D, E2F-DPs, Rb, and cyclin E are played by Cln3, SBF/MBF, Whi5, and Cln1,2 respectively[2, 52]. To date however, Whi5 was the only factor in yeast involved in maintaining the Start transcriptional program repressed before Start. Here, we have demonstrated that, as occurs in mammalian cells, yeast also uses multiple transcriptional repressors before Start, further extending the parallelism between yeast and mammalian cells. Due to the conserved regulatory pathways, understanding the interplay between Whi5 and Whi7 repressors in yeast could provide insight into how distinct Rb family proteins interweave in mammalian cell cycle control. The fact that an Rb family member is mutated in nearly all tumors underlies the importance of studying the role of these G1 repressors, and given the power of yeast as a model system, further investigation in Whi7 and Whi5 is warranted.

## Methods

**Strains and plasmids.** Yeast strains used in this work are shown in Supplementary Table 1. Centromeric plasmids pWHI7, pWHI7-NP, and pGAL1:WHI7 expressing HA-tagged wild-type Whi7, Whi7 protein mutated in all canonical CDK phosphorylation sites and wild-type Whi7 under the control of the *GAL1* promoter respectively, are a gift from Dr. M. Aldea[41]. Centromeric plasmid pGAL1:WHI5 expressing myc-tagged Whi5 under the control of the *GAL1* promoter is a gift from

Dr. C. Wittemberg. The *Spadh1:CLN2* centromeric plasmid express the Cln2 cyclin under the control of the mild *adh1* promoter from *S. pombe*. pSCB:lacZ express lacZ reporter under the control of a fragment from –507 to –367 from HO promoter bearing 3 SCB elements. Vector pYES2-GST and pYES2-GST-CDC4 and pYES2-GST-NES-CDC4 plasmids expressing wild-type Cdc4 or a Cdc4 protein fused to the nuclear export sequence from human PKI protein have been kindly provided by Dr. J. Benanti. pYES-Grr1ΔF-FLAG allows to overexpress a Grr1 protein lacking its F-box, such that it was still able to bind substrates without promoting its degradation, was a gift from Dr. D.P. Toczyski.

Cells were grown in exponentially conditions (below $1 \times 10^7$ cell mL$^{-1}$) at 25 °C in standard yeast extract-peptone-dextrose (YPD) medium or synthetic complete (SC) medium with 2% glucose, 2% raffinose or 2% galactose. Where indicated, cells were incubated in the presence of 15 µg mL$^{-1}$ nocodazole, 5 µg mL$^{-1}$ α-factor, or transferred to 37 °C.

**Protein stability assays**. To evaluate Whi7 and Whi5 stability, 100 µg mL$^{-1}$ cycloheximide was added to exponentially growing cells or cells incubated during 3 h at 37 °C when termosensitive mutants were used. Samples were harvested at the indicated times and protein decay was analyzed by western blot analysis. Bands were quantified with ImageQuant LAS 4000mini biomolecular imager (GE Healthcare). Two-three experiments with independent transformants were carried out for each strain. A representative experiment is shown.

**Chromatin immunoprecipitation analysis**. Approximately $5 \times 10^8$ cells were cross-linked by a 15 min incubation after the addition of formaldehyde to 1% (v/v) to the growth medium, followed by a 5 min incubation after the addition of glycine to 125 mM. Cells were washed and resuspended in 300 µL of lysis buffer (50 mM HEPES-KOH pH 7.9, 40 mM NaCl, 1 mM EDTA, 1% (v/v) Triton X-100, 0.1% (w/v) sodium deoxicholate, 1 mM PMSF, 1 mM benzamidine and Complete Mini protease inhibitor (Roche Diagnostics), and lysed by vortexing with glass beads for 30 min at 4 °C. Lysis buffer was supplemented to a final volume of 600 µL, chromatin was then fragmented by sonication and the sample was centrifuged at $13,400 \times g$ for 15 min. 20 µL from the supernatant was collected as a control of whole-cell extract (input) and the remaining was incubated with orbital rotation for 2 h at 4 °C with Dynabeads Protein G (Invitrogen) previously bound to an HA-probe (F-7) antibody (Santa Cruz Biotechnology Inc. Cat. No: SC-7392) or monoclonal GFP antibody (Roche Diagnostics, Cat. No: 11814460001). Beads were then washed 4 times in PBS (150 mM NaCl, 40 mM Na$_2$HPO$_4$, 10 mM NaH$_2$PO$_4$) containing 0.02% (v/v) Tween 20. Elution of bound protein was carried out twice with 40 µL of 50 mM Tris-HCl pH 8.0, 10 mM EDTA, 1% (w/v) SDS, by heating at 65 °C for 8 min. Cross-linking was reverted by overnight incubation at 65 °C with shaking. The eluted sample was digested for 90 min at 37 °C with 0.33 mg mL$^{-1}$ proteinase K and DNA was purified with the High Pure PCR product purification kit (Roche Diagnostics). Co-immunoprecipitated DNA was analyzed in triplicate by quantitative PCR in a DNA Engine Peltier Termal Cycler (Bio Rad) using the SYBR Premix Ex Taq Tli RNase H Plus Green with ROX (Takara). DNA analyzed are fragments from *CLN2*, *RNR1*, *FKS1*, and *MNN1* promoters and an intergenic region as a control. Whi7 or Whi5 binding values indicates the specific enrichment of the investigated promoter fragments in the immunoprecipitated sample compared to the whole-cell extract (input) using the intergenic region as a control, calculated with the ΔΔCT method. Values are relative to the no tag control strain (value of 1 equivalent to no specific enrichment).

**Gene expression analysis**. Approximately $2 \times 10^8$ cells were broken in a FastPrep Precellys24 (Bertin technologies) with glass beads in LETS buffer (0.1 M LiCl, 0.01 M EDTA, 0.01 M Tris-HCl pH 7.4, 0.2% SDS) with one volume of saturated phenol (pH 4.5). Cells were then centrifuged at $13,400 \times g$ for 5 min and the aqueous phase was extracted twice with one volume of phenol:chloroform (5:1) and one time with a volume of chloroform:isoamylic alcohol (24:1) and the RNA was precipitated overnight with one volume of 5 M lithium chloride at –80 °C. The precipitate was washed with 70% ethanol and resuspended in 30 µL of RNase-free water. RNA was quantified with NanoDrop 2000 Spectophotometer (Thermo Scientific). 5 µg of RNA were incubated with Turbo DNase (Ambion) and after DNase inactivation and incubation with oligo dT, cDNA was obtained with Improm-II Reverse Transcriptase and Recombinant RNasin (Promega) following the manufacturer instructions. The cDNA was analized by semiquantitative reverse transcription PCR (RT-PCR) or by quantitative RT-PCR as described above.

For β-galactosidase activity assays, ~$10^8$ cells were harvested, washed with water, and resuspended in 200 µL of Z buffer (60 mM Na$_2$HPO$_4$, 40 mM NaH$_2$PO$_4$·H$_2$O, 10 mM KCl, 1 mM MgSO$_4$, 0.1% β-mercaptoethanol, pH 7) containing 1 mg mL$^{-1}$ of zymolyase-20T. After 20 min of incubation at 30 °C, extracts were centrifuged and the supernatants were conserved. Different quantities of extracts (20–180 µL) were taken and Z buffer was added up to 900 µL. Reaction was started by the addition of 180 µL of 4 mg mL$^{-1}$ o-nitrophenyl-β-D-galactopiranoside. Samples are incubated at 30 °C until they become yellow colored. The reaction was stopped with 450 µL of Na$_2$CO$_3$ 1 M and the A$_{420}$ was determined. Activity is expressed in U µg$^{-1}$ of protein where 1U is defined as A$_{420} \times 10^3$ min$^{-1}$ of incubation. Protein concentration was determined by Bradford assay.

**Western blot analysis**. Approximately $10^8$ cells were collected, resuspended in 100 µL water and, after adding 100 µL 0.2 M NaOH, they were incubated for 5 min at room temperature. Cells were collected by centrifugation, resuspended in 50 µL sample buffer, and incubated for 5 min at 95 °C. Extracts were clarified by centrifugation, and equivalent amounts of protein were resolved in an SDS-PAGE gel and transferred onto a nitrocellulose membrane. The primary antibodies used were monoclonal anti-HA peroxidase 3F10 antibody (Roche Diagnostics, Cat. No: 12013819001) diluted 1:5000, monoclonal anti-c-myc 9E10 antibody (Roche Diagnostics, Cat. No: 1667149) diluted 1:5000, monoclonal anti-GFP (Roche Diagnostics, Cat. No: 11814460001) diluted 1:5000, monoclonal anti-FLAG M2 (Sigma-Aldrich, Cat. No: F3165) diluted 1:10000 and monoclonal anti Cdc2 p34 (PSTAIRE) (Santa Cruz Biotechnology Inc., Cat. No: SC-53) diluted 1:2000. Blots were developed with anti-mouse IgG and anti-rabbit IgG Horseradish Peroxidase conjugate (Thermo Fisher Scientific, Cat. No: 170-6516 or 31460 respectively) diluted 1:20000 using the Supersignal West Femto Maximum Sensitivity Substrate (Thermo Scientific). Bands were quantified with a ImageQuant$^{TM}$ LAS 4000mini biomolecular imager (GE Healthcare). Uncropped western blots are shown in Supplementary Figs. 9–15.

**Protein co-immunoprecipitation assays**. Approximately $5 \times 10^8$ cells expressing Grr1ΔF-flag, Swi4-myc, or Swi6-myc and either untagged or HA-epitope-tagged versions of Whi7 or Whi5 were resuspended in 100 µL of 50 mM Tris-HCl, pH 8, 250 mM NaCl, 5 mM EDTA, 0.1% Triton X-100, 1 mM PMSF, and 5 µg mL$^{-1}$ Complete mixture (Roche Applied Science) and broken with vigorous shaking in the presence of glass beads. Cellular debris was removed and the supernatant was clarified by centrifugation at $13,400 \times g$ for 5 min. Dynabeads Protein G magnetic beads (50 µL) were incubated with HA-probe (F-7) antibody (Santa Cruz Biotechnology Inc., Cat. No: SC-7392) for 20 min at room temperature and after extensive washing with phosphate-buffered saline containing Tween 0.02%, incubated with the cell extract for 20 min at room temperature. After washing beads, the immunoprecipitated proteins were eluted by boiling the beads for 5 min in loading buffer and analyzed by SDS-polyacrylamide gel electrophoresis followed by western blot analysis.

**Cell size analysis**. Cell size was analyzed in exponentially growing cells after brief sonication in a particle count and size Analyzer Z2 (Beckman Coulter). Graphs are the mobile average of histograms derived from values from at least six independent cultures.

**Data availability**. The data that support the findings of this study are available within the article and its Supplementary Information, and from the corresponding author on reasonable request.

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

## Acknowledgements

This work was supported by the Spanish Government and co-financed by ERDF from the European Union (grants number BFU2013-47503 and BFU2014-58429-P) and by the Generalitat Valenciana (grant number GVPROMETEO2016-123). E.M. is a recipient of a Predoctoral Fellowship from the Generalitat Valenciana.

## Author contributions

M.G.-A., E.M., and M.C.B. performed the experiments. M.G.-A., E.M., I.Q., and J.C.I. designed the experiments and interpreted the data. M.G.-A., I.Q. and J.C.I. wrote the manuscript. All authors read and discussed the manuscript.

## Additional information

**Competing interests:** The authors declare no competing financial interests

