## [Peer Review File · Nature Communications]

Reviewers' Comments:

Reviewer #1 (Remarks to the Author)

The article by Gomez-Alba et al describes the role of Whi7 as a transcriptional inhibitor during the cell cycle in budding yeast. The authors show that, similarly to Whi5, Whi7 binds G1/S-gene promoters and is phosphorylated during cell cycle entry but, as a key difference, Whi7 is unstable and degraded in a Cdc28- and SCF(Grr1)-dependent manner, pointing to Whi7 as a factor transmitting specific additional information to the G1/S regulon during entry into the cell cycle. The main conclusions of the article are correctly supported by the data and, as they provide new insights and unify G1/S control in yeast and mammalian cells, they will be interesting to a wide audience. However, there are some methodological and interpretation issues that should be properly addressed to strengthen the main conclusions of the article.

1. Assessment of a role in G1 is difficult in *cdc15*-release experiments. Cells become abnormally large during the arrest and growth requirements in subsequent G1 phase are largely suppressed. This is likely the reason why Whi5 is only shortly bound to the CLN2 promoter during the release of *cdc15*-arrested cells (Fig. 7B). In addition, the experimental *cdc15* setup to synchronize cells has important effects on Whi7 levels. Experiments in Fig 3C and 3D should be done in early-G1 elutriated and/or G1-arrested GAL-CLN3 *cln1,2* cells.

2. Fig 4C convincingly shows that Start execution is advanced in the *whi7 cln3* double mutant. Moreover, the authors nicely show that Whi7 overexpression causes a G1 delay in *cln3* cells (Fig 6), which are known to depend on redundant activities of Cln1,2 for Start execution and survival. Thus, as proposed by the authors, Whi7 would prevent the Cln1,2-transcriptional loop. However, Cln3 is thought as the initial trigger of this loop and, in consequence, Whi7 should also inhibit Cln3-dependent functions in the nucleus. This point needs to be discussed.

3. It would be very interesting to know if binding of Whi7 to the CLN2 promoter changes during G1. Due to reasons abovementioned, *cdc15*-arrested cells are not the best tool to answer this key question. These experiments should be conducted with early-G1, late-G1 and post-Start elutriated cells or, at least, in G1-arrested and released cells using the GAL-CLN3 *cln1,2* strain.

4. The authors do not observe important effects on cell size in Whi7-deficient (Fig S5) or overexpressing (Fig 7D) cells in a wild-type background, which contradict what was observed by Yahya et al (2014). Yet, clear effects are observed in *cln3*-deficient cells. This issue needs to be clarified.

5. Interesting differential roles of Whi5 and Whi7 are proposed by the authors in the discussion section. I would add here that Whi7, but not Whi5, levels are specifically increased in mother cells after more than 12 generations (Yiu et al 2008), which suggests a role for Whi7 in restraining cell cycle entry during replicative aging.

6. *cdc4*-arrested cells (Fig 2) show increased levels of Whi7 at time 0. These cells are supposed to arrest with high levels of Cln-Cdc28 activity, which would reinforce the notion proposed by the authors.

7. The title of the article should be more specific and refer to the role of Whi7 as an unstable transcriptional inhibitor of G1/S transcription.

Yiu et al. 2008. Pathways change in expression during replicative aging in *Saccharomyces cerevisiae*. *J Gerontol A Biol Sci Med Sci*. 63:21-34.

Reviewer #2 (Remarks to the Author)

This is a well-written and thoroughly studied manuscript. The experiments were carefully designed from both the genetic and the biochemical aspects to gather convincing experimental evidence for the authors to examine the critical role of Whi7 in governing Start progression in cell cycle regulation. The authors identified Whi7 as a ubiquitin substrate of SCF-Grr1 E3 ligase, and further showed that functions similarly to Whi5, Whi7 can suppress transcription to control Start progression. This study thus reveals an important new layer of transcriptional regulation through combinational actions of multiple transcriptional repressors to timely govern the Start transition in yeast cells. However, the following concerns should be addressed before its publication at Nature Communications.

1. Figure 1B, it will be critical for the authors to include Whi5 blot in this panel.
2. Figure 2, as presented in other figures, it will be nice for the authors to label each sub-panel A, B, C, D and E for future readers to easily follow.
3. Figure 3E, it will be important for the authors to gather biochemical evidence showing that WT, but not Whi7-NP, interacts with Grr1.
4. Figure S4, the authors showed that Cln3 is not involved in regulating Whi7. It will be critical for the authors to define the identity of cyclin that associates with Cdc28 to phosphorylate Whi7.
5. Figure 5A and 5E, it will be nice for the authors to side by side examine relative binding of Whi7 versus Whi5 on these gene promoters.
6. Figure 7A, in mammalian cells, Rb is phosphorylated by CDK at up to 13 Ser/Thr sites. The authors should explain why unlike Whi7 that migrates in both phosphorylated and non-phosphorylated forms, no slow migrating form, indicative of phosphorylated species of Whi5 was observed.
7. Figure 8, as the authors mentioned, Whi5 functions like Rb in mammalian cells, can the authors discuss or speculate the mammalian homologue of Whi7, is it also Rb-like protein such as p130 or p107?

Reviewer #3 (Remarks to the Author)

The manuscript titled "Cell cycle regulation of Whi7: new insights in Start progression" by Gomar-Alba and colleagues describes a study of the role and regulation of Whi7, a member of a family of proteins related by the presence of a GTB motif and involved in cell cycle regulation in budding yeast. The authors report that Whi7 accumulates periodically during the cell cycle as a consequence of its phosphorylation by G1 cyclin-associated CDK and targeting for destruction by the SCF-Grr1 ubiquitin ligase. They further show that Whi7, which has previously been described as a regulator of Cln3 translocation from the endoplasmic reticulum to the nucleus, also associates with SBF-regulated promoters like another GTB family member, Whi5. They find that binding to promoters is enhanced when Whi7 is stabilized by mutation of CDK phosphorylation sites or when cells overexpress Whi7 from the GAL1 promoter. That increase in promoter binding is associated with an extremely modest effect on cell cycle progression in otherwise wild type cells but is dramatically increased in CLN3-deficient cells. Consistent with the cellular phenotype, promoter binding of overexpressed Whi7 is enhanced and expression of SBF target genes is repressed in *cln3Δ* mutants. However, conversely, inactivation by the *whi7Δ* mutation has no discernable effect on the expression of SBF targets or the phenotype of otherwise wild type cells. Only when CLN3 is inactivated, compromising the robustness of Start, are its effects readily discernable. Furthermore, the effects of overexpression of Whi7 observed in wild type cells appear to be independent of SBF, suggesting they may be ectopic or affect the other reported function of Whi7. Finally, overexpressed Whi7 can complement a deficiency of WHI5 and deficiency of Whi7 enhances the small cell phenotype of a *whi5Δ* mutant, but only when CLN3 is also deficient.

The experiments described in this report are largely well designed and mostly wetted (see below for exceptions). The major conclusions regarding binding to SBF-regulated promoters and those concerning control of Whi7 protein stability are reasonably well established and novel (see below). It is satisfying that Whi7 binds to SBF target promoter since it is a member of a family of proteins

whose other members are known to bind the SBF and MBF transcription factors. However, the experiments addressing Whi7 function leave this reviewer uncertain of its role at SBF target promoters. The authors fail to identify a phenotypic consequence of inactivation of WHI7 in otherwise wild type cells. Observing a phenotype associated with inactivation requires mutational perturbation of Start control. Similarly, cellular phenotypes associated with hyperaccumulation of Whi7 are very modest unless other mutations are present. Finally, the later phenotypes appear, in some cases, to be associated with functions unrelated to SBF-regulated promoters, perhaps via the reported role of Whi7 in regulating Cln3 translocation to the nucleus. Consequently, the importance of promoter binding and cell cycle-regulated protein stability are unclear, thereby compromising the impact of the study. Specific issues concerning the experiments and their conclusions are detailed below.

Specific comments:

1. The data in Figures 6 and 7, showing the lack of an effect of *whi7Δ* in wild type cells, but a relatively robust effect in *cln3Δ* and *swi4Δ* or *swi6Δ* mutant cells, shows that Whi7 is important when G1 is substantially perturbed. Furthermore, overexpression of Whi7 has a significant phenotypic effect under the same conditions. However, it remains unclear whether Whi7 plays a significant role in cells and the phenotypes may just as well reflect a role acquired under compromising conditions. The fact that Whi7 has already been reported to play a role in the regulation of Cln3 independent of its role in transcriptional regulation confuses the interpretation of these results and raises concerns regarding role at the promoters. The paper establishes that it can play a role at promoters but that its role is unimportant in wild type cells.

2. The authors should be more careful in their interpretation of the experiments, especially those in Figures 6 and 7, when discussing the role of Whi7 in the discussion. The experiment in Figure 7D shows that Whi7 can suppress *whi5Δ* phenotype when it is overexpressed but that function is not apparent at wild type level of Whi7. It must be surmised from the fact that Figure 7C has been performed in a *cln3Δ* mutant, that the modest effect of *whi7Δ* seen in a *whi5Δ* mutant is only seen when CLN3 is deleted. In fact, that same effect is seen in a *cln3Δ* mutant when WHI5 is present suggesting that it is not resulting from the loss of complementation of the *whi5Δ* mutant but from an independent role that is revealed only in the absence of CLN3.

3. Figure 4D showing the effect of *whi7Δ* on cell cycle progression should be presented as a complete cell cycle. Especially because the cell cycle proceed at different rates in the two mutants. This presentation is difficult to interpret and may be deceptive in terms of the effect of the mutant. The authors should have data from the entire time course shown in experiments in 4B and 4C.

4. The binding at the promoter appears to be significant and enhanced by the non-phosphorylatable mutant of Whi7. Furthermore, that binding does appear to depend, in part, upon Swi4. However, the data in Figure 5F, showing of binding to Swi4 and Swi6 by coimmunoprecipitation, is unconvincing. Unless more convincing data can be produced, it does little the advance the story being presented.

5. The stabilization of the Whi7 protein appears to be quite incomplete in all of the SCF-related mutants except for *grr1Δ* and, to some extent, *cdc53Δ*, which show a more robust result. However, even there, the instability is significant. Does this imply that there are multiple pathways for destruction? This possibility should be discussed.

6. The demonstration of phosphatase sensitivity of Whi7 seems to have started with largely unphosphorylated Whi7. That experiment should be repeated with Whi7 that is significantly phosphorylated (60' after release from a *cdc15-ts* arrest, for example) to demonstrate phosphatase sensitivity.

7. There are some bothersome differences between results in figures. For example, Whi7 protein accumulation is shown to be periodic in Figure 1B but does not appear to be periodic at all in Figure 7B yet the experiments appear to have been done using the same *cdc15* mutant release protocol. Figure 3C also shows the Whi7 protein depleted at a *cdc15-ts* arrest but Figure 7B does not. What is the explanation for this? Is the protein reliably periodic or is it not?

RESPONSE TO REFEREES

Manuscript: **NCOMMS-16-28169**. "Cell cycle regulation of Whi7: new insights in Start progression"

Reviewer #1:

The article by Gomez-Alba et al describes the role of Whi7 as a transcriptional inhibitor during the cell cycle in budding yeast. The authors show that, similarly to Whi5, Whi7 binds G1/S-gene promoters and is phosphorylated during cell cycle entry but, as a key difference, Whi7 is unstable and degraded in a Cdc28- and SCF(Grr1)-dependent manner, pointing to Whi7 as a factor transmitting specific additional information to the G1/S regulon during entry into the cell cycle. The main conclusions of the article are correctly supported by the data and, as they provide new insights and unify G1/S control in yeast and mammalian cells, they will be interesting to a wide audience. However, there are some methodological and interpretation issues that should be properly addressed to strengthen the main conclusions of the article.

1) Assessment of a role in G1 is difficult in cdc15-release experiments. Cells become abnormally large during the arrest and growth requirements in subsequent G1 phase are largely suppressed. This is likely the reason why Whi5 is only shortly bound to the CLN2 promoter during the release of cdc15-arrested cells (Fig. 7B). In addition, the experimental cdc15 setup to synchronize cells has important effects on Whi7 levels. Experiments in Fig 3C and 3D should be done in early-G1 elutriated and/or G1-arrested GAL-CLN3 *cln1,2* cells.

We have carried out a shut-off experiment in cells arrested in G1 by depletion of Cln cyclins. The result is included in new Figure 3C and confirmed that Whi7 remains in G1 as a non-phosphorylated stable protein.

2) Fig 4C convincingly shows that Start execution is advanced in the whi7 *cln3* double mutant. Moreover, the authors nicely show that Whi7 overexpression causes a G1 delay in *cln3* cells (Fig 6), which are known to depend on redundant activities of Cln1,2 for Start execution and survival. Thus, as proposed by the authors, Whi7 would prevent the Cln1,2-transcriptional loop. However, Cln3 is thought as the initial trigger of this loop and, in consequence, Whi7 should also inhibit Cln3-dependent functions in the nucleus. This point needs to be discussed.

In the model it was reflected the idea that both Cln3-dependent initial trigger of SBF transcription as well as Cln1,2- dependent transcriptional activation are restricted by Whi7. We have now stated this in the text in page 12.

3) It would be very interesting to know if binding of Whi7 to the CLN2 promoter changes during G1. Due to reasons abovementioned, cdc15-arrested cells are not the best tool to answer this key question. These experiments should be conducted with early-G1, late-G1 and post-Start elutriated cells or, at least, in G1-arrested and released cells using the GAL-CLN3 *cln1,2* strain.

We have analysed Whi7 binding to *CLN2* promoter in synchronized cells by G1-arrest and release using the *GAL1:CLN3 cln1 cln2* strain. Data is included in new Figure 5B. No binding is observed in the arrested cells and the results confirmed the transient binding to *CLN2* promoter before Start activation once cells are released from the arrest.

4) The authors do not observe important effects on cell size in *Whi7*-deficient (Fig S5) or overexpressing (Fig 7D) cells in a wild-type background, which contradict what was observed by Yahya et al (2014). Yet, clear effects are observed in *cln3*-deficient cells. This issue needs to be clarified.

To clarify this point, we have analysed the budding index in mid-log asynchronous cultures and have found a reduction in the percent of unbudded cells in *whi7* mutant (18.9%) compared to wild type (24.3%), and, conversely, an increase in the percent of unbudded cells when *WHI7* is overexpressed with a *GAL1:WHI7* plasmid (49.4%) compared to control (32.2%). These data are consistent with those described in Yahya et al (2014) and reflect a shortening or lengthening of G1 phase when *Whi7* is inactivated or overexpressed, respectively. These data are now stated in the text in page 7 and page 9.

As it is shown in the manuscript, this effect does not result in an alteration in cell size distribution in asynchronous cultures in the *whi7* mutant. To rule out a background effect we carried out cell size analysis in the same strain used by Yahya, confirming the lack of significant effect in cell size distribution (now this graph is included in Supplementary Figure 5). This fact could reflect some kind of compensation in other cell cycle stages. On the other hand, Yahya et al. reported that *whi7* cells bud at a size of approximately 48-49 fL, whereas wild type cells bud at a size of approximately 54 fL, so cell size at bud is reduced approximately a 10%. Maybe, this mild reduction is not reflected in a cell size distribution analysis. Note that in the case of *whi5* mutation, which indeed manifests a clear effect in cell size distribution in mid-log cultures, a more important reduction in cell size at budding (approximately 30%) is observed (Ferrezuelo et al. Nat. Comm. 3:1012 (2012)).

5) Interesting differential roles of *Whi5* and *Whi7* are proposed by the authors in the discussion section. I would add here that *Whi7*, but not *Whi5*, levels are specifically increased in mother cells after more than 12 generations (Yiu et al 2008), which suggests a role for *Whi7* in restraining cell cycle entry during replicative aging.

We weren't aware of this observation, which is really very interesting. We have included it in the Discussion section in page 13.

6) *cdc4*-arrested cells (Fig 2) show increased levels of *Whi7* at time 0. These cells are supposed to arrest with high levels of *Cln-Cdc28* activity, which would reinforce the notion proposed by the authors.

We have not pursued the significance of the difference at time 0. Inactivation of *Cdc4* slightly stabilizes *Whi7*, which could explain this observation, or it could be caused by the cell cycle oscillation in *Whi7* protein level.

7) The title of the article should be more specific and refer to the role of *Whi7* as an unstable transcriptional inhibitor of G1/S transcription.

In our opinion the title is appropriate. '*Cell-cycle regulation*' involves not only changes in phosphorylation status and protein stability but also change in *Whi7* association to promoters, and although our results show a new function of *Whi7* as a transcriptional inhibitor at Start, we think that taking into account the previous reported function in the cytosol, it is better to use a general expression like '*new insights in Start progression*' that to focus only on transcription.

RESPONSE TO REFEREES

Manuscript: **NCOMMS-16-28169**. "Cell cycle regulation of Whi7: new insights in Start progression"

Reviewer #2:

This is a well-written and thoroughly studied manuscript. The experiments were carefully designed from both the genetic and the biochemical aspects to gather convincing experimental evidence for the authors to examine the critical role of Whi7 in governing Start progression in cell cycle regulation. The authors identified Whi7 as a ubiquitin substrate of SCF-Grr1 E3 ligase, and further showed that functions similarly to Whi5, Whi7 can suppress transcription to control Start progression. This study thus reveals an important new layer of transcriptional regulation through combinational actions of multiple transcriptional repressors to timely govern the Start transition in yeast cells. However, the following concerns should be addressed before its publication at Nature Communications.

1) Figure 1B, it will be critical for the authors to include Whi5 blot in this panel.

We have carried out a new experiment, similar to that in Figure 1, in the new Figure 7B in which it can be seen in parallel the Whi7 and Whi5 proteins in a synchronized culture (see answer to point 6).

2) Figure 2, as presented in other figures, it will be nice for the authors to label each sub-panel A, B, C, D and E for future readers to easily follow.

Done.

3) Figure 3E, it will be important for the authors to gather biochemical evidence showing that WT, but not Whi7-NP, interacts with Grr1.

We have carried out a co-IP experiment with Grr1-flag that demonstrates that Grr1 specifically interacts with the phosphorylated form of Whi7. The result is shown in new Figure 2F.

4) Figure S4, the authors showed that Cln3 is not involved in regulating Whi7. It will be critical for the authors to define the identity of cyclin that associates with Cdc28 to phosphorylate Whi7.

We stated in the original manuscript that the instability of Whi7 in a *cln3* mutant in cell cycle stages where Clbs are the only existing cyclins, suggested that Clbs are involved in Whi7 degradation. We have now carried out an experiment to directly prove this analysing Whi7 stability in a *clb1clb2clb3clb4* mutant strain. The result is shown in new Supplementary Figure 4B and revealed that these cyclins are indeed required for proper Whi7 degradation. On the other hand, Whi7 is unstable in *cdc4* mutant strain, which has low Clb-Cdc28 activities due to the accumulation of the CKI Sic1; we have now demonstrated that inactivation of Cln1 and Cln2 in these cells results in Whi7 stabilization (Supplementary Figure 4C). All our observations consistently indicate a redundant role for the distinct cyclin-Cdc28 activities in targeting Whi7 for degradation.

5) Figure 5A and 5E, it will be nice for the authors to side by side examine relative binding of Whi7 versus Whi5 on these gene promoters.

We have analysed in parallel binding of Whi7 and Whi5 to different promoters. The results are shown in new Figure 7D.

6) Figure 7A, in mammalian cells, Rb is phosphorylated by CDK at up to 13 Ser/Thr sites. The authors should explain why unlike Whi7 that migrates in both phosphorylated and non-phosphorylated forms, no slow migrating form, indicative of phosphorylated species of Whi5 was observed.

Changes in Whi5 electrophoretic mobility associated to cell cycle are barely detected (figure 4 in de Bruin et al 2004, ref 11) or not detected (figure 5 in Constanzo et al. 2004, ref 10). This could be due to the existence of the reported Whi5 phosphorylation by non-CDK kinases; in fact, mutation of CDK phosphorylation sites does not alter Whi5 migration (figure 2 in Wagner et al. 2009, ref 17). In new Figure 7B, it can be indeed detected a slight alteration of Whi5 migration in 30-60 min samples similar to that described in de Bruin et al 2004; interestingly, this change correlates with the maximal hyperphosphorylation of Whi7.

7) Figure 8, as the authors mentioned, Whi5 functions like Rb in mammalian cells, can the authors discuss or speculate the mammalian homologue of Whi7, is it also Rb-like protein such as p130 or p107?

p130 seems to be the most important pocket protein during quiescence, whereas RB and p107 are found at higher levels in cycling cells. A recent study has not revealed a function for Whi5,7 proteins in establishing quiescence in yeast but in the resume of proliferation (Miles et al. 2016, ref 27). On the other hand, it has been reported that Whi7 levels increase in mother cells suggesting a possible role in cellular senescence (Yiu et al 2008 J. Gerontology ABSMS 63:21). Although quiescence and senescence are different processes, could Whi7 play, as p130 in mammalian, a specific role in establishing a non-proliferative state? Our studies has been done in cycling cells, we haven't analysed Whi7/5 binding to promoters in quiescent or senescence cells. We feel for now more comfortable avoiding to correlate Whi7 with one particular pocket protein, which would be highly speculative, and prefer to just remark that the combined action of more than one Start transcriptional repressors also occurs in yeast. As commented in Discussion, we hope that the experimental feasibility offered by yeast could help to understand how is the interplay among different repressors and hence to better understand the function of pocket proteins in mammalian cells. Maybe then, it would be possible to correlate Whi7 and Whi5 with specific pocket proteins.

RESPONSE TO REFEREES

Manuscript: **NCOMMS-16-28169**. "Cell cycle regulation of *Whi7*: new insights in Start progression"

Reviewer #3:

The manuscript titled "Cell cycle regulation of *Whi7*: new insights in Start progression" by Gomar-Alba and colleagues describes a study of the role and regulation of *Whi7*, a member of a family of proteins related by the presence of a GTB motif and involved in cell cycle regulation in budding yeast. The authors report that *Whi7* accumulates periodically during the cell cycle as a consequence of its phosphorylation by G1 cyclin-associated CDK and targeting for destruction by the SCF-Grr1 ubiquitin ligase. They further show that *Whi7*, which has previously been described as a regulator of *Cln3* translocation from the endoplasmic reticulum to the nucleus, also associates with SBF-regulated promoters like another GTB family member, *Whi5*. They find that binding to promoters is enhanced when *Whi7* is stabilized by mutation of CDK phosphorylation sites or when cells overexpress *Whi7* from the *GAL1* promoter. That increase in promoter binding is associated with an extremely modest effect on cell cycle progression in otherwise wild type cells but is dramatically increased in *CLN3*-deficient cells. Consistent with the cellular phenotype, promoter binding of overexpressed *Whi7* is enhanced and expression of SBF target genes is repressed in *cln3Δ* mutants. However, conversely, inactivation by the *whi7Δ* mutation has no discernable effect on the expression of SBF targets or the phenotype of otherwise wild type cells. Only when *CLN3* is inactivated, compromising the robustness of Start, are its effects readily discernable. Furthermore, the effects of overexpression of *Whi7* observed in wild type cells appear to be independent of SBF, suggesting they may be ectopic or affect the other reported function of *Whi7*. Finally, overexpressed *Whi7* can complement a deficiency of *WHI5* and deficiency of *Whi7* enhances the small cell phenotype of a *whi5Δ* mutant, but only when *CLN3* is also deficient.

The experiments described in this report are largely well designed and mostly weted (see below for exceptions). The major conclusions regarding binding to SBF-regulated promoters and those concerning control of *Whi7* protein stability are reasonably well established and novel (see below). It is satisfying that *Whi7* binds to SBF target promoter since it is a member of a family of proteins whose other members are known to bind the SBF and MBF transcription factors. However, the experiments addressing *Whi7* function leave this reviewer uncertain of its role at SBF target promoters. The authors fail to identify a phenotypic consequence of inactivation of *WHI7* in otherwise wild type cells. Observing a phenotype associated with inactivation requires mutational perturbation of Start control. Similarly, cellular phenotypes associated with hyperaccumulation of *Whi7* are very modest unless other mutations are present. Finally, the later phenotypes appear, in some cases, to be associated with functions unrelated to SBF-regulated promoters, perhaps via the reported role of *Whi7* in regulating *Cln3* translocation to the nucleus. Consequently, the importance of promoter binding and cell cycle-regulated protein stability are unclear, thereby compromising the impact of the study. Specific issues concerning the experiments and their conclusions are detailed below.

The Reviewer is concerned about the apparent lack of phenotype of *whi7* mutation in wild type cells. Functional redundancy is a common trait in cell cycle regulation; therefore inactivation of some regulators has no important effects on its own. Cyclins are a good example since many of them could be eliminated without apparent phenotype. However, this is not the case of *Whi7* since inactivation of *Whi7* does have an effect in wild type cells. Yahya et al. 2014 reported that it causes a reduction in cell size at budding. Importantly, we have now determined budding index in mid-log asynchronous cultures and we have found a reduction in the percent of unbudded cells in *whi7* mutant (18.9%) compared to wild type (24.3%), and, conversely, an increase in the percent of unbudded cells when *WHI7* is overexpressed with a *GAL1:WHI7* plasmid (49.4%) compared to control vector (32.2%). These data are consistent with those described in Yahya et al (2014) and reflect a shortening or

lengthening of G1 phase when Whi7 is inactivated or overexpressed, respectively. These data are now stated in the text in page 7 and page 9.

As it is shown in the manuscript, this effect does not result in an alteration in cell size distribution in asynchronous cultures in the *whi7* mutant. To rule out a background effect we carried out cell size analysis in the same strain used by Yahya, confirming the lack of significant effect in cell size distribution (now this graph is included in Supplementary Figure 5). This fact could reflect some kind of compensation in other cell cycle stages or a limitation of the technique. Yahya et al. reported that *whi7* cells bud at a size of approximately 48-49 fL, whereas wild type bud at a size of approximately 54 fL, so cell size at bud is reduced approximately a 10%. Maybe, this mild reduction is not reflected in a cell size distribution analysis. Note that in the case of *whi5* mutation, which indeed manifests a clear effect in cell size distribution in mid-log cultures, a more important reduction in cell size at budding (approximately 30%) is observed (Ferrezuelo et al. Nat. Comm. 3:1012 (2012)).

Specific comments:

1) The data in Figures 6 and 7, showing the lack of an effect of *whi7Δ* in wild type cells, but a relatively robust effect in *cln3Δ* and *swi4Δ* or *swi6Δ* mutant cells, shows that Whi7 is important when G1 is substantially perturbed. Furthermore, overexpression of Whi7 has a significant phenotypic effect under the same conditions. However, it remains unclear whether Whi7 plays a significant role in cells and the phenotypes may just as well reflect a role acquired under compromising conditions. The fact that Whi7 has already been reported to play a role in the regulation of Cln3 independent of its role in transcriptional regulation confuses the interpretation of these results and raises concerns regarding role at the promoters. The paper establishes that it can play a role at promoters but that its role is unimportant in wild type cells.

The lack of effect of *whi7* mutation in wild type cells and whether Whi7 plays a significant role has been commented above. We would like to add that as commented in answer to point 2, Whi7 seems to be less abundant than Whi5, which could explain the weaker phenotype of *whi7* mutation compared to *whi5*.

In any case, we believe that the use of a *cln3* background along the manuscript, which is remarked by the Reviewer as a weakness, is indeed not only valuable but necessary to investigate Whi7 function. It is impossible to evaluate Whi7 transcriptional repressor functions in a *CLN3* background since phenotype could have been caused by the Whi7 regulation of Cln3 RE-association described in Yahya et al. 2014. A *cln3* background eliminates that contribution. Our results unequivocally demonstrate that Whi7 plays a repressive role in Start progression that is different from that described in Yahya et al. 2014. Indeed, Yahya et al. results could be reinterpreted in the light of the results presented here. The fact that Whi7 overexpression is lethal in a *cln3* mutant background (i.e. Whi7-dependent Cln3 regulation off, Whi7-dependent SBF repression on) but not in a SBF mutant background (i.e. Whi7-dependent Cln3 regulation on, Whi7-dependent SBF repression off), suggests that Start repression by Whi7 is mediated more strongly through transcriptional repression of SBF.

2) The authors should be more careful in their interpretation of the experiments, especially those in Figures 6 and 7, when discussing the role of Whi7 in the discussion. The experiment in Figure 7D shows that Whi7 can suppress *whi5Δ* phenotype when it is overexpressed but that function is not apparent at wild type level of Whi7. It must be surmised from the fact that Figure 7C has been performed in a *cln3Δ* mutant, that the modest effect of *whi7Δ* seen in a *whi5Δ* mutant is only seen when *CLN3* is deleted. In fact, that same effect is seen in a *cln3Δ* mutant when *WHI5* is present suggesting that it is not resulting from the loss of complementation of the *whi5Δ* mutant but from an independent role that is revealed only in the absence of *CLN3*.

original Figure 7C, now Figure 7E.

Reviewer states: 'it must be surmised from the fact that Figure 7C has been performed in a *cln3* mutant, that the modest effect of *whi7* seen in a *whi5* mutant is only seen when *CLN3* is deleted'. We have to remark that cell size analysis of *whi7 whi5* was already shown in the original Supplementary Figure 5 and commented in the last paragraph of the Results section (page 10) and in Discussion (second paragraph, page 13). The fact of the lack of changes in cell size distribution in mid-log asynchronous cultures of *whi7* mutant has been commented above.

Maybe there is some confusion about the interpretation of the result in original Figure 7C (now 7E). *Whi7* inactivation causes a cell size reduction in *cln3* cells. *Whi5* inactivation also causes a cell size reduction of *cln3* cells. Simultaneous *Whi7* and *Whi5* inactivation have additive effects showing a bigger cell size reduction of *cln3* cells. This indicates that *Whi7* and *Whi5* act in parallel branches to control cell size in *cln3* cells. We think this is what Figure 7E means and in this way was commented in Results (page 10) and Discussion (page 13).

Related to this, we have carried out an analysis of cell size in cells overexpressing *WHI7* and/or *WHI5*. The result is shown in new Figure 7F and indicates that o.e.*Whi7* has an additive effect to o.e.*Whi5*, further demonstrating that both proteins act in parallel branches to control cell cycle.

original Figure 7D, now Figure 7G.

We have pursued comparison of *Whi7* and *Whi5*. We tagged both proteins with the same cassette and we observed that *Whi7* is less abundant than *Whi5*. This is now included in Figure 7C. This could help to explain why overexpression of *Whi7* is required to compensate the absence of *Whi5*.

3) Figure 4D showing the effect of *whi7Δ* on cell cycle progression should be presented as a complete cell cycle. Especially because the cell cycle proceeds at different rates in the two mutants. This presentation is difficult to interpret and may be deceptive in terms of the effect of the mutant. The authors should have data from the entire time course shown in experiments in 4B and 4C.

In the experiment in Figure 4 we used a 5-min time scale in order to have a greater resolution in the analysis of G1 phase. We understand Reviewer's concern and because of that we have now included a complete cell cycle experiment in Supplementary Figure 6. The result is the same to that shown in Figure 4. It reveals a specific shortening of G1 phase and not a general advance in cell cycle progression in the absence of *Whi7*. Note that a 10-min instead of 5-min interval between samples was used in Fig S6. The *cdc15* strain used to manifest a cytokinesis delay that makes it impossible in most of the cases to determine a budding index, except from the appearance of the bud in the first cycle. Cell cycle progression is more accurately evaluated by the analysis of Sic1 protein level, in particular when focusing in G1 phase.

4) The binding at the promoter appears to be significant and enhanced by the non-phosphorylatable mutant of *Whi7*. Furthermore, that binding does appear to depend, in part, upon *Swi4*. However, the data in Figure 5F, showing of binding to *Swi4* and *Swi6* by coimmunoprecipitation, is unconvincing. Unless more convincing data can be produced, it does little to advance the story being presented.

Swi6 co-IP experiment have been improved (new Figure 5F). Additionally, we have included parallel Co-IP assays for *Whi5* to better evaluate the results. A specific enrichment in the IP samples is observed and importantly, the amount of *Swi4* and *Swi6* co-purified with *Whi7* is in the range of that observed with *Whi5*.

5) The stabilization of the Whi7 protein appears to be quite incomplete in all of the SCF-related mutants except for *grr1Δ* and, to some extent, *cdc53Δ*, which show a more robust result. However, even there, the instability is significant. Does this imply that there are multiple pathways for destruction? This possibility should be discussed.

It is true that stabilization is not complete, but this is a common observation in other targets. For instance, see Cln2 shut-offs in Barral et al 1995, *Genes&Dev* 9:399 (figure 4), Willems et al 1996, *Cell* 86:543 (figure 4), Patton et al 1998, *Genes&Dev* 12:692 (figure 6), Kishi and Yamao 1998, *JCS* 111:3655 (figure 5). In the case of *ts* mutants it can be envisaged the possibility that a remaining activity could mediate degradation, but this certainly cannot apply for *grr1* deletion mutant. Because of that, as the Reviewer states, other pathways should contribute to Whi7 degradation. We had this in mind and stated that Whi7 is mainly or mostly degraded by SCFGrr1 to avoid overinterpretation. It has been reported that Cln3 and Cln2, which are mainly degraded by Grr1, could also be targeted for degradation by Cdc4 (Landry et al. 2012, *Plos Genet* 8; Quilis and Igual 2017, *FEBS OpenBio* 7:74). This could be also the case for Whi7 since we detected that Cdc4 inactivation slightly stabilizes Whi7. Alternatively, Whi7 could have a higher intrinsic instability. To clarify this point, we have now rewritten the second paragraph in page 11.

6) The demonstration of phosphatase sensitivity of Whi7 seems to have started with largely unphosphorylated Whi7. That experiment should be repeated with Whi7 that is significantly phosphorylated (60' after release from a *cdc15-ts* arrest, for example) to demonstrate phosphatase sensitivity.

A new phosphatase assay with more phosphorylated Whi7 is now included in Supplementary Figure 1.

7) There are some bothersome differences between results in figures. For example, Whi7 protein accumulation is shown to be periodic in Figure 1B but does not appear to be periodic at all in Figure 7B yet the experiments appear to have been done using the same *cdc15* mutant release protocol. Figure 3C also shows the Whi7 protein depleted at a *cdc15-ts* arrest but Figure 7B does not. What is the explanation for this? Is the protein reliably periodic or is it not?

A fluctuation in Whi7 protein level indeed existed in original Figure 7B with a 2.9-fold increase in 60 min sample referred to 0 min sample (note that 0 min sample was overloaded as seen in the loading control protein). This is a consistent result seen in all performed experiments, although quantitative differences in the amplitude of oscillation could be observed, most probably depending on the robustness of synchronization. In any case, we have improved the experiment in Figure 7B; new result further confirms Whi7 oscillation (4-fold more protein at 60 min than at 0 min).

Reviewers' Comments:

Reviewer #1:

Remarks to the Author:

The authors have performed the suggested experiments and obtained clear results that sustain further the main conclusions of the article. In addition, minor discrepancies observed in the whi7 mutant regarding effects on cell size have been properly addressed. In my view, the article is now acceptable for publication in Nat. Commun. As the revised version convincingly shows that Whi7 is not associated to the CLN2 promoter in G1-arrested cells, I would just suggest the authors to specify that Whi7 associates to G1/S gene promoters in "late G1"... in the abstract and in page 8 (second paragraph).

Reviewer #2:

Remarks to the Author:

The authors have addressed most of the raised concerns during this round of revision.

Reviewer #3:

Remarks to the Author:

The manuscript by Igual and colleagues is substantially improved both by addition of new data and alterations to the text. In addition, the authors pointed out correctly that I had misinterpreted one set of experiments shown in Figure 7. Consequently, I am satisfied with the demonstration that Whi7 binds to promoters, collaborates with Whi5 to regulate G1/S transcription and that its abundance is regulated via Grr1 in a manner that depends upon phosphorylation by cyclin dependent protein kinases. The cell cycle regulation and effect on budding and cell size are much clearer in this iteration of the manuscript. I agree with Reviewer 1 that the title is non-committal and would be improved by making a positive statement about the conclusions of the study.

One concern about the new data: Supplementary Figure 4C purportedly shows that inactivation of Cln1 and Cln2 in a *cdc4-ts* arrest leads to the stabilization of Whi7. However, this result is confusing because in the *cln1 cln2* mutant phosphorylated Whi7 appears very unstable and a small fraction of Whi7 that is unphosphorylated appears very stable. If Cln1 and Cln2 are required for phosphorylation in G1, it is unexpected that Wh7 is phosphorylated at 0'. Furthermore, the phosphorylated protein is less stable than the phosphorylated form observed when Cln1 and Cln2 are present.

RESPONSE TO REFEREES

Manuscript: **NCOMMS-16-28169**. "Cell cycle regulation of Whi7: new insights in Start progression"

Reviewer #3:

The manuscript by Igual and colleagues is substantially improved both by addition of new data and alterations to the text. In addition, the authors pointed out correctly that I had misinterpreted one set of experiments shown in Figure 7. Consequently, I am satisfied with the demonstration that Whi7 binds to promoters, collaborates with Whi5 to regulate G1/S transcription and that its abundance is regulated via Grr1 in a manner that depends upon phosphorylation by cyclin dependent protein kinases. The cell cycle regulation and effect on budding and cell size are much clearer in this iteration of the manuscript. I agree with Reviewer 1 that the title is non-committal and would be improved by making a positive statement about the conclusions of the study.

Following Reviewer's suggestion we have modified the title. Now states: "Whi7 is a new cell-cycle unstable repressor of the Start transcriptional program"

One concern about the new data: Supplementary Figure 4C purportedly shows that inactivation of Cln1 and Cln2 in a *cdc4-ts* arrest leads to the stabilization of Whi7. However, this result is confusing because in the *cln1 cln2* mutant phosphorylated Whi7 appears very unstable and a small fraction of Whi7 that is unphosphorylated appears very stable.

As Reviewer states, the residual phosphorylated Whi7 at time 0 is highly unstable and the observed stable protein accumulates as the non-phosphorylatable form. This is expected since we are demonstrating that Whi7 phosphorylation targets the protein to degradation; therefore stable Whi7 accumulates as a non-phosphorylated protein (Figures 3C, 3D, 3E). Note that what we do in these experiments is to short-circuit new Whi7 phosphorylation but the original phosphorylated protein present at time 0 is expected to be degraded.

If Cln1 and Cln2 are required for phosphorylation in G1, it is unexpected that Whi7 is phosphorylated at 0'.

We also want to note that it is not surprising that a residual phosphorylated Whi7 was observed at time 0 in *cln1 cln2*. This is probably due to residual CDK activities: it has to be considered that Cln3 cyclin is present and that some Clb-Cdc28 kinases could escape from inhibition by the CKI Sic1.

Furthermore, the phosphorylated protein is less stable than the phosphorylated form observed when Cln1 and Cln2 are present.

This is a dynamic situation and in the presence of Cln1 Cln2 there is a high rate of Whi7 phosphorylation that provides an important continuous input of new phosphorylated protein. Because of that, the rate of disappearance of phosphorylated Whi7 cannot be directly compared.

Reviewers' Comments:

Reviewer #3:

Remarks to the Author:

The authors have largely addressed my questions. This was a relatively minor point which I did not expect to be asked further about.

The new title is better than the prior one. However, it would be more appropriately stated as "Whi7 is a new unstable repressor of the Start transcriptional program" or "Whi7 is a new unstable repressor of the cell cycle Start transcriptional program" if they prefer to have cell cycle in the title.